# Visible light-driven efficient palladium catalyst turnover in oxidative transformations within confined frameworks

Jiawei Li[1,2], Liuqing He[2], Qiong Liu[3], Yanwei Ren [1✉] & Huanfeng Jiang [1✉]

Palladium catalyst turnover by reoxidation of a low-valent Pd species dominates the proceeding of an efficient oxidative transformation, but the state-of-the-art catalysis approaches still have great challenges from the perspectives of high efficiency, atom-economy and environmental-friendliness. Herein, we report a new strategy for addressing Pd reoxidation problem by the fabrication of spatially proximate Ir[III] photocatalyst and Pd[II] catalyst into metal-organic framework (MOF), affording MOFs based Pd/photoredox catalysts UiO-67-Ir-PdX$_2$ (X = OAc, TFA), which are systematically evaluated using three representative Pd-catalyzed oxidation reactions. Owing to the stabilization of single-site Pd and Ir catalysts by MOFs framework as well as the proximity of them favoring fast electron transfer, UiO-67-Ir-PdX$_2$, under visible light, exhibits up to 25 times of Pd catalyst turnover number than the existing catalysis systems. Mechanism investigations theoretically corroborate the capability of MOFs based Pd/photoredox catalysis to regulate the competitive processes of Pd$^0$ aggregation and reoxidation in Pd-catalyzed oxidation reactions.

[1] Key Laboratory of Functional Molecular Engineering of Guangdong Province, School of Chemistry and Chemical Engineering, South China University of Technology, 510641 Guangzhou, People's Republic of China. [2] College of Chemistry and Chemical Engineering, Central South University, 410083 Changsha, People's Republic of China. [3] Institute of Analysis, Guangdong Academy of Sciences (China National Analytical Center, Guangzhou), 510070 Guangzhou, People's Republic of China. ✉email: renyw@scut.edu.cn; jianghf@scut.edu.cn

P alladium-catalyzed oxidation reactions, such as Wacker-type oxidations, diacetoxylation of 1,3-butadiene, oxidative esterification of methacrolein, etc., are responsible for the production of billion pounds of fine chemicals each year. The development of streamlined Pd-catalyzed oxidation systems that are efficiently, economically, and ecologically advantageous would be highly desirable for their practical applications in the coming resource-scarce era. Generally, the competitive processes of $Pd^0$ aggregation and reoxidation in oxidative transformation dominate the proceeding of an efficient catalytic cycle (Fig. 1a). Compared with the thermodynamic favored $Pd^0$ aggregation to Pd nanoparticles (NPs) or Pd black, the reoxidation process is kinetically more challenging. Therefore, the Pd catalyst turnover by reoxidation of the $Pd^0$ species is often the rate-limiting step in the oxidation reaction[1].

The state-of-the-art approaches toward the Pd reoxidation can be classified into three main routes. The molecular oxygen ($O_2$) as low-cost, abundant and environmental benign oxidant, fulfills the requirement for green chemistry, but direct reoxidation of low-valent Pd species by atmosphere $O_2$ is usually kinetically unfavored[2–7]. Instead, stoichiometric oxidants, such as $Cu^{II}$, $Ag^I$, and benzoquinone (BQ), have been employed as the most frequently utilized approach for addressing this problem[8–10]. However, the inclusion of excess undesired oxidants gives rise to low atom economy and deleterious organic waste. One alternative approach is the substitution of stoichiometric oxidants to catalytic amount of electron transfer mediators (ETMs) or photocatalyst with $O_2$ as the terminal oxidant[1,11–20]. It serves as a milder manner to complete the Pd reoxidation via stepwise electron transfer between Pd, ETMs, or photocatalyst and $O_2$. Nevertheless, compared with stoichiometric oxidants, the catalytic amounts of ETMs or photocatalyst in solution reduce the chance to interact with $Pd^0$ species, and the overall Pd catalyst turnover

efficiency is thus unsatisfactory. For these reactions, the Pd reoxidation still remains problematic in current Pd-catalyzed oxidation reactions, and high Pd loadings (e.g., 10 mol%) are inevitably involved in most catalytic systems.

Theoretically, low Pd catalyst consumption is sufficient to support an efficient oxidative transformation when the reoxidation rate of $Pd^0$ species exceeds its aggregation rate, and this requires an effective strategy to both accelerate the reoxidation and restrain the $Pd^0$ aggregation processes in the catalytic cycle. In this context, metal-organic frameworks (MOFs) as highly porous and tunable platform would be a judicious selection[21–26]. The merger of photocatalyst and transition metal catalysts into MOFs have been reported recently, and this elegant methodology has demonstrated their successful applications in photocatalytic water splitting[27–29], $CO_2$ reduction[30–32], and organic transformations[33–36] by promoting electron transfer and stabilizing active intermediates. For example, a Ru-Pt@UiO-67 MOF assembly allows a facile arrangement of the photosensitizer and the reduction catalyst with close spatial proximity to promote the electron transfer between them, and thus leading to a significantly improved hydrogen evolution activity[29]. However, to our knowledge, MOF-based Pd/photoredox composite has never been explored for Pd catalyst turnover in oxidation reactions. We envision MOFs framework can offer a promising platform for the regulation of the competitive $Pd^0$ aggregation and reoxidation processes (Fig. 1b). Moreover, the well-defined structures of MOFs can provide facile opportunity to reveal the stepwise electron transfer process between Pd, photocatalyst, and $O_2$, which further gives insight for the elucidation of the Pd reoxidation pathway.

Herein, we present two recyclable MOFs based Pd/photoredox catalysts UiO-67-Ir-$PdX_2$ ($X = OAc$, TFA) through the hierachical integration of poly(pyridine)–$Ir^{III}$ complex photocatalyst and $Pd^{II}$ into a porous MOF backbone. The femtosecond transient absorption spectroscopy (fs-TAS) track in real time the electron transfer process between the well-ordered Pd and Ir centers, which demonstrates the ultrafast Pd reoxidation within the framework. An in-depth comparison between this MOFs based Pd/photoredox catalysis and the aforementioned approaches for Pd reoxidation were systematically investigated and evaluated using three representative Pd-catalyzed oxidation reactions, such as C−H alkenylation of 2-phenylphenol, decarboxylative coupling of allylic alcohols and acetoxypalladation of alkynes with alkenes. Owing to the stabilization of single-site Pd and Ir catalysts by the MOFs framework as well as the high local concentration and the proximity of them favoring fast electron transfer, the MOFs based Pd/photoredox catalysis represents the combination of high-turnover Pd catalyst with atom-economy, sustainability, and environmental-friendliness towards Pd-catalyzed oxidation reactions.

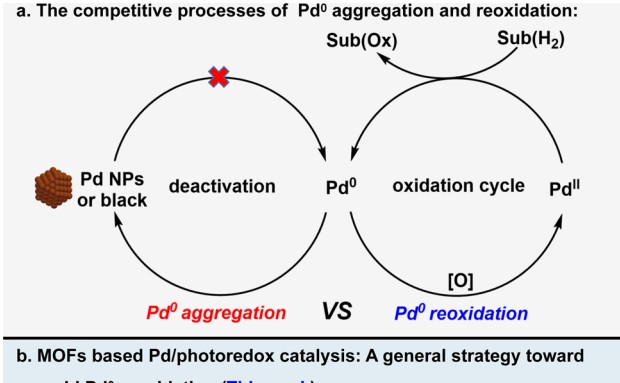

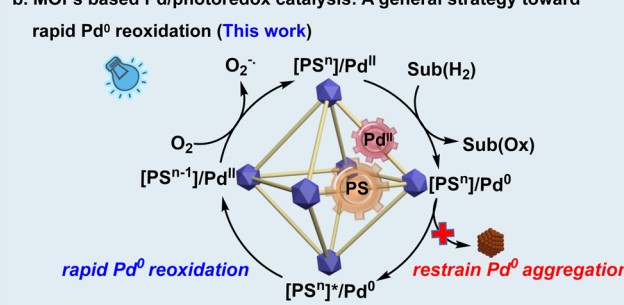

**Fig. 1 Schematic illustration of Pd reoxidation. a** The competitive processes of $Pd^0$ aggregation and reoxidation in Pd-catalyzed oxidation reactions, the existing strategies toward $Pd^0$ reoxidation suffer from low Pd turnover number, high-pressure $O_2$ atmosphere, organic waste, low atom economy, separation difficulty. **b** MOFs based Pd/photoredox catalysis for $Pd^0$ reoxidation with high Pd turnover efficiency, atom economy, environmental-friendliness, and recyclability.

## Results

**Synthesis and characterizations of MOFs catalysts.** The poly(pyridine)-$Ir^{III}$ complexes have won profound reputation as photosensitizers (PS) because they allow the visible light-mediated charge separation with long-lived excited states, and can be facilely modulated through the ligand design to improve their photophysical properties, which made them extensively studied in varieties of photocatalytic transformations[37]. Recently, the immobilization of poly(pyridine)-$Ir^{III}$ complexes into MOF frameworks has been demonstrated to be an efficient strategy to accelerate the SET process and improve the overall photocatalytic performances[38–41]. Herein, for the first time, we attempted the integration of poly(pyridine)-$Ir^{III}$ complex and $Pd^{II}$ catalyst into MOF strut to serve as a promising new strategy to resolve the

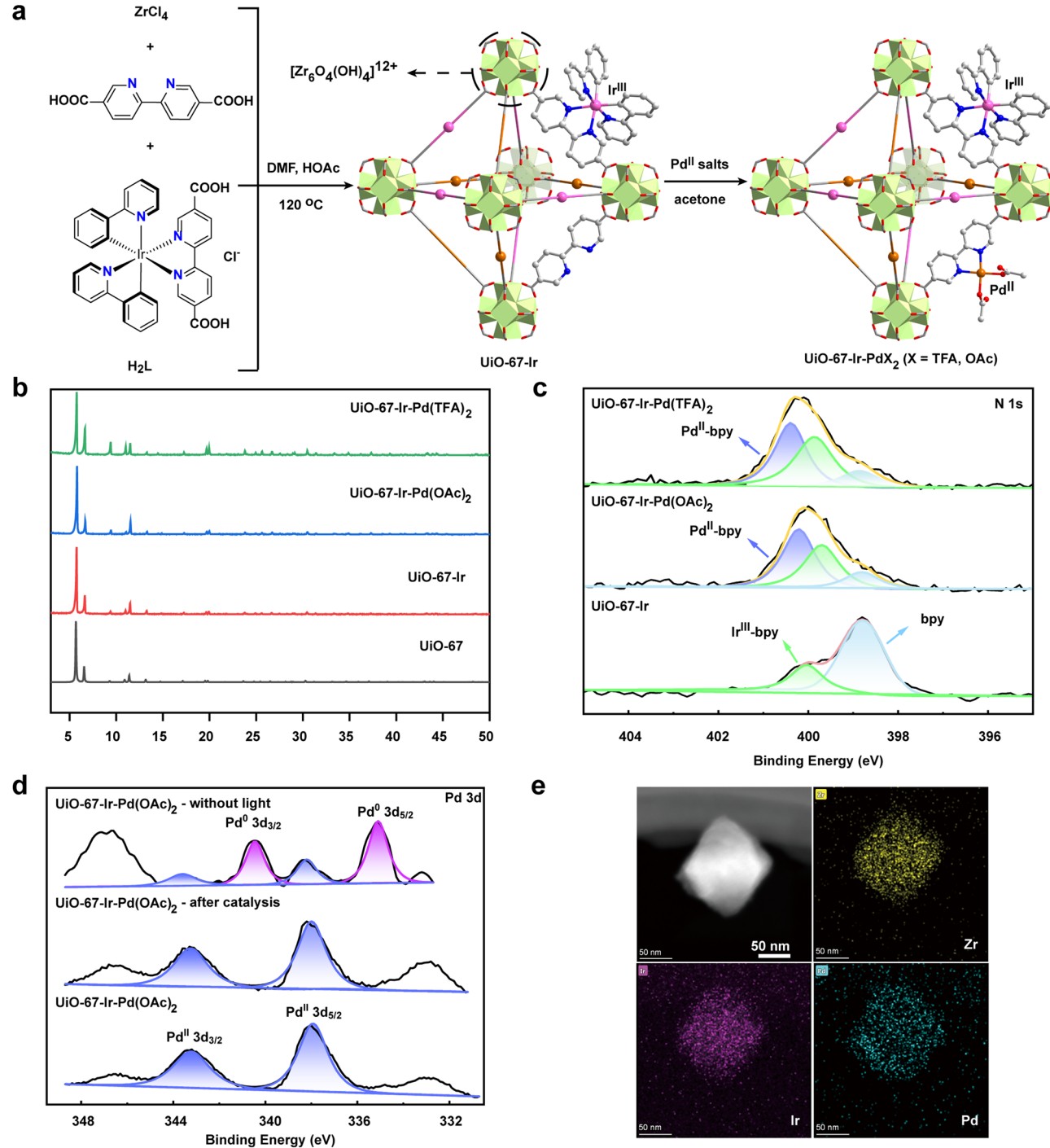

**Fig. 2 Structure and characteristics of UiO-67-Ir-Pd$X_2$ ($X$ = OAc, TFA). a** Syntheses of UiO-67-Ir-Pd$X_2$ via pre-functionalization and post-synthetic modification. **b** PXRD patterns of UiO-67-Ir-Pd(OAc)$_2$, UiO-67-Ir-Pd(TFA)$_2$, UiO-67-Ir, and UiO-67. **c** XPS spectra of the N 1s region of UiO-67-Ir-Pd$X_2$ and UiO-67-Ir. **d** XPS spectra of the Pd 3d region for different UiO-67-Ir-Pd(OAc)$_2$ catalysts. **e** TEM and elemental mapping images of UiO-67-Ir-Pd(OAc)$_2$.

fundamental Pd reoxidation problem in Pd-catalyzed oxidation reactions.

In this work, the bipyridine functionalized Zr-based MOF UiO-67 was selected as carrier since it not only possess the features of porosity, stability, and tunability, but also can be modified its structure via the bipyridine docking sites to anchor metal complexes with proximate distance (< 1 nm)[42–45]. As shown in Fig. 2a, the Ir[III] complex PS and Pd[II] catalyst were judiciously incorporated into UiO-67 framework in two steps combining pre-functionalization and post-synthetic modification

to afford UiO-67-Ir-Pd$X_2$ ($X$ = OAc, TFA). Specifically, the dicarboxylate ligand [Ir(ppy)$_2$(dcbpy)]Cl [**H$_2$L**; where ppy = 2-phenylpyridine, bpy = 2,2′-bipyridine-5,5′-dicarboxylate] was initially synthesized by the direct treatment of Ir$_2$(ppy)$_4$Cl$_2$ with bpy in CH$_3$OH at 70 °C overnight (Supplementary Fig. 1). The UiO-67-Ir was then synthesized by mixing ZrCl$_4$, bpy and **H$_2$L** in DMF at 120 °C for 24 h with HOAc as the modulator. Powder X-ray diffraction (PXRD) pattern confirmed the purity and crystallinity of UiO-67-Ir, which matches well with that of UiO-67 (Fig. 2b). [1]H NMR analysis of the D$_3$PO$_4$-digested UiO-67-Ir

gave a bpy/**L**/HOAc ratio of 7.14: 1: 1.91 (Supplementary Fig. 4), suggesting an approximate empirical formula of UiO-67-Ir as $[Zr_6O_4(OH)_4bpy_{4.71}(\mathbf{L})_{0.66}(OAc)_{1.26}]$, which was further evidenced by the thermogravimetric analysis with a weight loss of 62.17 % (theoretical: 62.85 %) (Supplementary Fig. 5). The diffuse reflectance UV/vis spectra of UiO-67-Ir showed prominent adsorption in the visible light region due to the incorporation of $Ir^{III}$ complex PS within the framework, apparently different from that of the pure UiO-67 (Supplementary Fig. 10). The X-ray photoelectron spectroscopy (XPS) analyses were also conducted to ensure the embedding of $Ir^{III}$ complex into the MOF. As shown in Fig. 2c, there are two N 1s peaks of binding energy (BE) in UiO-67-Ir, signifying two kinds of coordination environment of N atoms. The BE of N 1s peak at 398.8 eV can be ascribed to the N atoms in free bpy ligand, similar to that of UiO-67-bpy[20,46], while the one located at 400 eV is attributed to the $Ir^{III}$-coordinated N atoms (Supplementary Fig. 13). In addition, the $4f_{7/2}$ and $4f_{5/2}$ Ir peaks appeared at 61.88 and 64.86 eV confirmed +3 oxidation state of Ir in UiO-67-Ir (Supplementary Fig. 12)[47]. The $N_2$ adsorption at 77 K indicated a type I isotherm with Brunauer-Emmett-Teller (BET) surface area of 962 $m^2/g$ (Supplementary Fig. 9), comparable to that of previously reported Ir-UiO MOF[27]. The scanning electron microscopy (SEM) and transmission electron microscopy (TEM) images of UiO-67-Ir showed octahedral morphology with a diameter of 50-200 nm (Supplementary Fig. 3), which are beneficial to facilitate the photosensitization[48], the exposure of more accessible active sites[22] and the effective adsorption of oxygen[49].

The post-synthetic metalation of UiO-67-Ir with $Pd(OAc)_2$ [or $Pd(TFA)_2$] was conducted in acetone at 50 °C for 24 h to afford UiO-67-Ir-Pd$X_2$ ($X$ = OAc, TFA) whose PXRD patterns and morphology remained unchanged (Fig. 2b and Supplementary Fig. 8). Inductively coupled plasma mass spectrometry (ICP-MS) analyses of UiO-67-Ir-Pd(OAc)$_2$ and UiO-67-Ir-Pd(TFA)$_2$ implied Zr/Ir/Pd molar ratios of 6: 0.63: 0.62 and 6: 0.67: 0.65, respectively. The reduced $N_2$ adsorption amounts and calculated BET values of UiO-67-Ir-Pd$X_2$ ($X$ = OAc, TFA) can be explained by the incorporation of $Pd^{II}$ salts into the framework (Supplementary Fig. 9). The TEM images together with elemental mapping images of UiO-67-Ir-Pd$X_2$ ($X$ = OAc, TFA) confirmed the uniform distribution of Zr, Ir, and Pd over the octahedral crystals (Fig. 2e). The successful coordination of $Pd^{II}$ ion to the free bpy moieties in UiO-67-Ir was evidenced by XPS analyses. The characteristic BE peaks of Pd $3d_{5/2}$ at 337.9, 338.3 eV and Pd $3d_{3/2}$ at 343.2, 343.6 eV for UiO-67-Ir-Pd(OAc)$_2$ and UiO-67-Ir-Pd(TFA)$_2$ (Fig. 2d and Supplementary Fig. 20), respectively, unambiguously suggesting the inclusion of $Pd^{II}$ salts within the framework[20]. Moreover, for both UiO-67-Ir-Pd(OAc)$_2$ and UiO-67-Ir-Pd(TFA)$_2$, there are three kinds of N 1 s BE peaks located at 398.8, 399.7, 400.2 and 398.9, 399.9, 400.4 eV, respectively (Fig. 2c), demonstrating the coexistence of bpy, $Ir^{III}$-bpy and $Pd^{II}$-bpy species within the framework (Supplementary Fig. 13). The $Ir^{III}$ oxidation state in UiO-67-Ir-Pd$X_2$ ($X$ = OAc, TFA) was also confirmed unchanged during the post-synthetic metalation process by XPS (Supplementary Fig. 12).

To further determine the coordination environments and oxidation states of Pd and Ir in UiO-67-Ir-Pd$X_2$ ($X$ = OAc, TFA), X-ray absorption fine structure was carried out. X-ray absorption near-edge structure (XANES) spectroscopy of various standard species suggested the $Pd^{II}$ and $Ir^{III}$ oxidation states in UiO-67-Ir-Pd$X_2$ ($X$ = OAc, TFA), as with their homogeneous Pd and Ir complexes (Fig. 3a, c). Fitting of the extended X-ray absorption fine structure (EXAFS) data for UiO-67-Ir-Pd(OAc)$_2$ at Pd K-edge reveals that the coordination number of Pd is about 4, signifying the coordination to two N atoms from the bpy ligand and two O atoms from two OAc$^-$ groups with average Pd−N/O

bonds length of 2.00 Å (Fig. 3b, Supplementary Table 1). This result matches well with the above XPS data of N atom in UiO-67-Ir-Pd(OAc)$_2$ (Fig. 2c). In addition, no obvious Pd-Pd metallic interaction was observed in either UiO-67-Ir-Pd(OAc)$_2$ or UiO-67-Ir-Pd(TFA)$_2$, ruling out the formation of Pd NPs. The EXAFS characteristic of UiO-67-Ir-Pd(OAc)$_2$ at Ir $L_3$-edge is consistent with the reported crystal structures of corresponding $Ir^{III}$ complex $[Ir(bpy)(ppy)_2](PF_6)$ with almost the same coordination environment (Supplementary Fig. 16a)[50]. To be specific, EXAFS results showed that Ir center adopts an octahedral geometry by the coordination to four N atoms from one bpy and two ppy ligands, and two C atoms from two ppy ligands with average Ir−N$_{ppy}$ bond length of 2.05 Å, Ir−C$_{ppy}$ bond length of 1.97 Å, and Ir−N$_{bpy}$ bond length of 2.12 Å (Fig. 3d, Supplementary Table 2). Each path for the corresponding Ir-C/N bonds has been shown in Supplementary Fig. 16c in detail. UiO-67-Ir-Pd(TFA)$_2$ features a similar octahedral $Ir^{III}$ center with one bpy and two ppy and four-coordination $Pd^{II}$(bpy)(TFA)$_2$ moiety (Supplementary Figs. 14, 16b). These results corroborate the successful integration of single-site $Ir^{III}$ PS and $Pd^{II}$ catalysts with well-defined coordination geometry into the UiO-67 framework.

**Catalytic performances**. With the well-prepared UiO-67-Ir-Pd$X_2$ in hand, we started to investigate the capability of MOFs based Pd/photoredox catalysis as a general strategy to resolve the Pd reoxidation problem in three representative Pd-catalyzed oxidation reactions, such as C−H alkenylation of 2-phenylphenol[51], decarboxylative coupling of allylic alcohols[52] and acetoxypalladation of alkynes with alkenes[53], which suffered from high Pd consumptions (5–10 mol%) and (sub)stoichiometric external oxidants (BQ, $Ag^I$ and $Cu^{II}$ salts, respectively) at elevated temperatures. A systematic evaluation and comparison between MOFs based Pd/photoredox catalysis and the existing three main approaches toward Pd reoxidation were conducted using these above reactions.

Taking the Pd-catalyzed decarboxylative coupling of allylic alcohols as an example, when atmosphere $O_2$ was used as the sole oxidant, the reaction between **1a** and **2a** in the presence Pd(TFA)$_2$ (5 mol%) at 100 °C was totally quenched (Table 1, entry 1). This result indicates the kinetically unfavorable $Pd^0$ reoxidation process by $O_2$[14], and the thermodynamically favored $Pd^0$ aggregation to Pd black shuts down the reaction. For comparison, the utilization of stoichiometric amounts of $Ag_2CO_3$ (100 mol%) could facilitate the $Pd^0$ reoxidation process and gave corresponding product **3a** with a yield of 85 % (Table 1, entry 2). The utilization of catalytic amounts of photocatalyst Ir(ppy)$_3$ (1 mol%) and $O_2$, in substitution of stoichiometric $Ag_2CO_3$, serves as a milder manner to complete the $Pd^0$ reoxidation process, but the diluted photoexcited $[Ir^{III}]^*$ species in solution results in largely decreased $Pd^0$ reoxidation efficiency (Table 1, entries 3–5). It is noteworthy that although stoichiometric oxidants or the combination of photocatalyst and $O_2$ can promote the $Pd^0$ reoxidation, the competitive process of $Pd^0$ aggregation in the catalytic cycle occurs inevitably. Therefore, high Pd consumption is indispensable for these catalytic systems.

In sharp contrast, the MOFs based Pd/photoredox catalyst, UiO-67-Ir-Pd(TFA)$_2$, merging single-site $Ir^{III}$ PS and $Pd^{II}$ catalyst in a porous MOF backbone exhibits remarkably enhanced catalytic performances from different points of views. Firstly, as shown in Fig. 4a, the reaction between **1a** and **2a** in the presence of UiO-67-Ir-Pd(TFA)$_2$ proceeded smoothly under varying $Pd^{II}$ catalyst loadings and the catalyst consumption can be as low as 0.25 mol%, while the reduction of the Pd loadings in Pd/stoichiometric-oxidant and Pd/photoredox systems result in apparent lose in the catalytic activity. This catalytic data implied

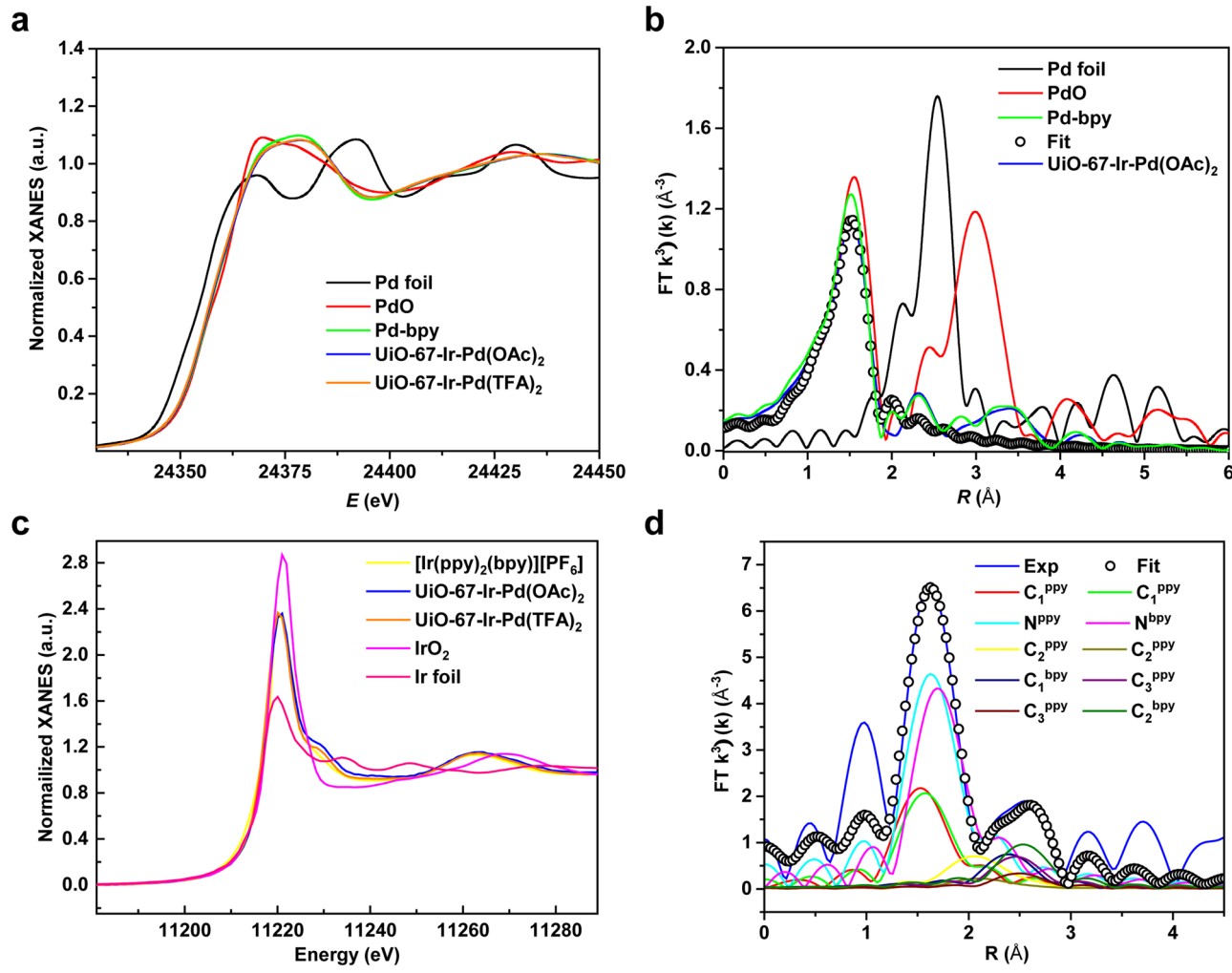

**Fig. 3 XANES and EXAFS characteristics of UiO-67-Ir-Pd$X_2$ ($X$ = OAc, TFA). a** Normalized XANES features of Pd foil (black), PdO (red), Pd-bpy (green), UiO-67-Ir-Pd(OAc)$_2$ (blue) and UiO-67-Ir-Pd(TFA)$_2$ (orange). **b** EXAFS spectra in R space at the Pd K-edge adsorption of Pd foil (black), PdO (red), Pd-bpy (green), and UiO-67-Ir-Pd(OAc)$_2$ (blue). **c** Normalized XANES features of Ir foil (pink), IrO$_2$ (magenta), [Ir(bpy)(ppy)$_2$](PF$_6$) (yellow), UiO-67-Ir-Pd(OAc)$_2$ (blue) and UiO-67-Ir-Pd(TFA)$_2$ (orange). **d** EXAFS spectrum (bule line) and fit (black circles) in R space at the Ir L$_3$-edge adsorption of UiO-67-Ir-Pd(OAc)$_2$.

a remarkably improved Pd catalyst TON of UiO-67-Ir-Pd(TFA)$_2$ with up to 11 times higher than the aforementioned catalytic systems (Fig. 4d). Secondly, the Pd catalyst turnover efficiency of MOFs based Pd/photoredox catalysis and the existing approaches are also distinctly different. Since O$_2$ as the sole oxidant failed to promote the Pd$^0$ reoxidation process of this reaction, the Pd catalyst turnover is completely inhibited under this condition. For Pd/stoichiometric-oxidant system, excess oxidant in solution can interact with the Pd$^0$ species sufficiently, leading to a moderate Pd catalyst TOF value (Fig. 4d). In terms of the Pd/photoredox system, catalytic amounts of [Ir$^{III}$]* in solution reduce the chance to undergo efficient electron transfer with Pd$^0$ species, and the overall Pd catalyst TOF is thus remarkably reduced. Thanks to the spatially proximate Ir$^{III}$ photocatalyst and Pd$^{II}$ catalyst anchored on the MOF, the in situ formed Pd$^0$ species could undergo rapid electron transfer to [Ir$^{III}$]* even at a low catalyst loading, giving prominently elevated Pd catalyst TOF up to 26 times over homogeneous counterparts (Fig. 4d). This significant differences in Pd catalyst turnover efficiency can also be stemmed from the stability of their corresponding catalytic systems. As shown in Fig. 4b, the kinetic rate of the homogeneous Pd/photoredox system decreases apparently along with the reaction time, while the MOFs based Pd/photoredox system, on the

contrary, remain highly active during the three consecutive catalytic runs. Moreover, no detectable Pd$^0$ NPs were found in the recovered UiO-67-Ir-Pd(TFA)$_2$ catalyst even after five consecutive runs (Supplementary Fig. 22). These results implied that the single-site Pd$^{II}$ catalyst and Ir$^{III}$ PS within MOF may eliminate the multimolecular deactivation pathway and suppress in situ formed Pd$^0$ aggregation process that are usually encountered in homogeneous systems, thus making MOF system possess high stability and efficiency during the catalytic cycle. Benefit from its stability and heterogeneity, UiO-67-Pd(TFA)$_2$ could be recovered by simple centrifugation and used for five cycles without loss of catalytic activity (Fig. 4c). In contrast, the catalysts in homogeneous systems were difficult to recover, and could not be reused for extra catalytic run. Last but not the least, the reaction in the presence of UiO-67-Ir-Pd(TFA)$_2$ can be conducted with atmosphere O$_2$ at room temperature under visible light, representing greatly improved atom economy and environmental-friendliness compared to the Pd/stoichiometric-oxidant system which could only be proceeded with the prerequisite of excess oxidants at 100 °C.

Control experiments proposed that the reaction was quenched when performed in dark even at elevated temperature or in the absence of O$_2$ (Table 1, entries 8 and 9). Besides, UiO-67-

**Table 1 Comparison of the homogeneous Pd/O₂, Pd/stoichiometric-oxidant, Pd/photoredox and MOFs based Pd/photoredox systems for the decarboxylative coupling reaction[a].**

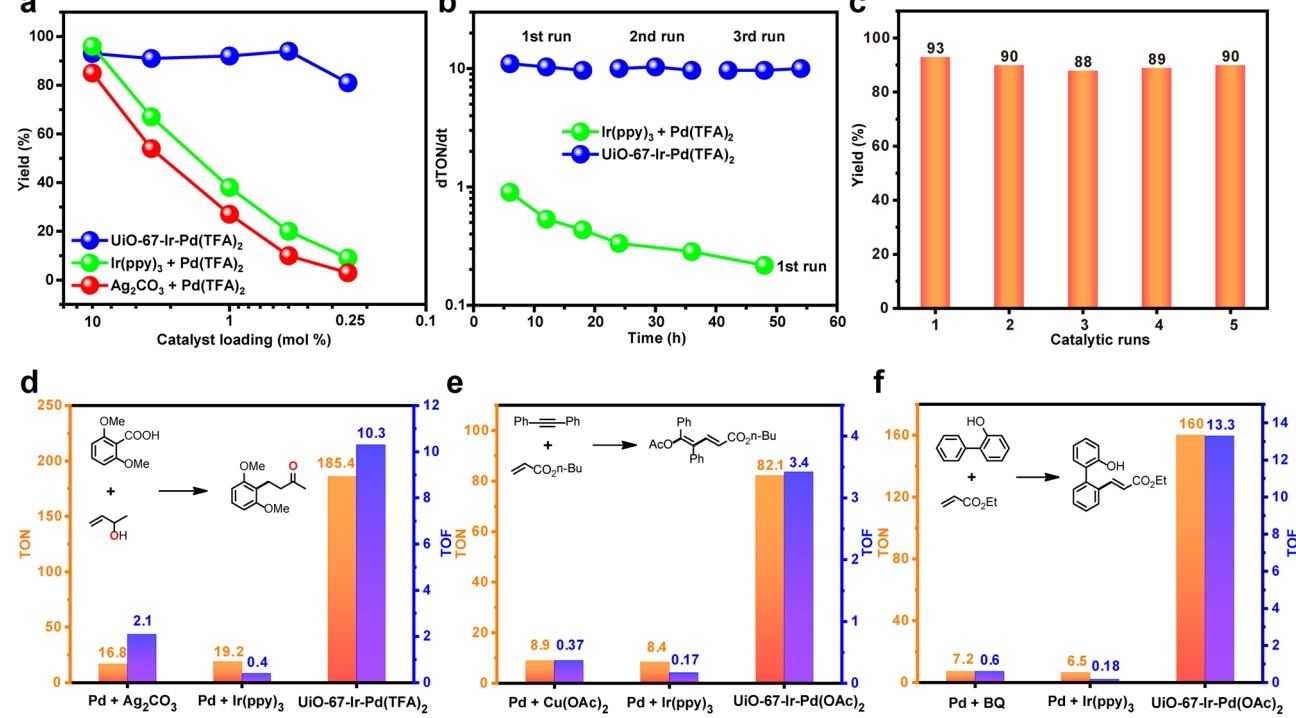

| Entry | Catalytic system | t (h) | T (°C) | Yield (%)[b] | TON[c] | TOF[d] |
|---|---|---|---|---|---|---|
| *Homogeneous Pd/O₂ system* | | | | | | |
| 1 | Pd(TFA)₂ (5 mol%)/O₂ (1 atm) | 8 | 100 | n.d. | – | – |
| *Homogeneous Pd/stoichiometric oxidants system* | | | | | | |
| 2 | Pd(TFA)₂ (5 mol%)/Ag₂CO₃ (100 mol%) | 8 | 100 | 85 | 16.8 | 2.1 |
| *Homogeneous Pd/photoredox system[e]* | | | | | | |
| 3 | Pd(TFA)₂ (5 mol%)/Ir(ppy)₃ (1 mol%) | 18 | 25 | 56 | 11.2 | 0.62 |
| 4 | Pd(TFA)₂ (0.5 mol%)/Ir(ppy)₃ (1 mol%) | 48 | 25 | 20 | 39.8 | 0.83 |
| 5 | Pd(TFA)₂ (5 mol%)/Ir(ppy)₃ (1 mol%) | 48 | 25 | 96 | 19.2 | 0.4 |
| *MOFs based Pd/photoredox system[e]* | | | | | | |
| 6 | UiO-67-Ir-Pd(TFA)₂ [Pd (0.5 mol%)] | 18 | 25 | 93 | 185.4 | 10.3 |
| 7 | UiO-67-Ir-Pd(TFA)₂ [Pd (0.5 mol%)] | 12 | 25 | 64 | 127.2 | 10.6 |
| 8[f] | UiO-67-Ir-Pd(TFA)₂ [Pd (0.5 mol%)] | 18 | 50 | <5 | – | – |
| 9[g] | UiO-67-Ir-Pd(TFA)₂ [Pd (0.5 mol%)] | 18 | 25 | Trace | – | – |
| 10 | UiO-67-Pd(TFA)₂ [Pd (0.5 mol%)] | 18 | 25 | Trace | – | – |
| 11 | UiO-67-Ir [Ir (0.5 mol%)] | 18 | 25 | n.d. | – | – |
| 12 | UiO-67-bpy (1 mol%) /Pd(TFA)₂ (5 mol%)/Ir(ppy)₃ (1 mol%) | 18 | 25 | 51 | 10.2 | 0.57 |

[a]Reaction conditions: **1a** (0.5 mmol), **2a** (0.6 mmol), solvent (3 mL).
[b]Yields are determined by NMR with CH₂Br₂ as internal standard.
[c]TON (turnover number): moles of product per mole of catalyst.
[d]TOF (turnover frequency): moles of product per mole of catalyst per hour.
[e]O₂ atmosphere (1 atm), blue LED (40 W).
[f]In dark.
[g]Under N₂.

**Fig. 4 Activity and recyclability of the selected catalysts. a** Plots of yields for **3a** vs Pd[II] catalyst loading of UiO-67-Pd(TFA)₂ and the homogeneous systems under their respective optimized reaction conditions. **b** the kinetic rates of UiO-67-Ir-Pd(TFA)₂ and homogenous mixed system. **c** recyclability tests of UiO-67-Ir-Pd(TFA)₂ in the decarboxylative coupling between **1a** and **2a**. TON and TOF values of UiO-67-Pd*X*₂ and the homogeneous systems under their respective optimized reaction conditions for **d** decarboxylative coupling, **e** acetoxypalladation of alkynes with alkenes, **f** C−H alkenylation of 2-phenylphenol.

$Pd(TFA)_2$ (Supplementary Fig. 6) without the $Ir^{III}$ PS, or UiO-67-Ir without the $Pd^{II}$ catalyst in MOF framework could not initiate the reaction (Table 1, entries 10 and 11). These findings suggest the irreplaceable role of visible light, $Ir^{III}$ PS, and $O_2$ for the Pd reoxidation process. The physical mixture of UiO-67-bpy, $Pd(TFA)_2$ and $Ir(ppy)_3$ only gave similar catalytic result to the homogeneous Pd/photoredox system, implying the significant importance of the spatially proximate $Ir^{III}$ and $Pd^{II}$ catalysts within the framework (Table 1, entry 12). It is worth noting that UiO-67-Ir-$Pd(TFA)_2$ was stable under the photocatalytic reaction conditions as illustrated by the retention of PXRD pattern for the recovered UiO-67-Ir-$Pd(TFA)_2$ (Supplementary Fig. 19) as well as the leaching of <0.5% Pd and <0.3 % Ir as determined by ICP-MS. The slight changes in the PXRD relative reflex intensities of MOFs catalyst before and after use may be due to the partial structural distortion of the catalyst surface after catalysis. XPS measurements of the catalyst after catalysis showed the oxidation states of Pd (+2) and Ir (+3) remain unchanged (Supplementary Figs. 20, 21). The TEM image of the recovered UiO-67-Ir-$Pd(TFA)_2$ after five consecutive runs showed that the octahedral morphology was slightly distorted, while the elemental mapping images indicate that the Zr, Pd, Ir elements are still uniformly distributed over the framework (Supplementary Fig. 22).

The MOFs based Pd/photoredox catalysis as a general strategy could be further applied to Pd-catalyzed C−H alkenylation of 2-phenylphenol and acetoxypalladation of alkyne with alkene, which also showed overwhelming merits in terms of high efficiency, atom economy, recyclability, and environmental-friendliness over aforementioned other catalytic systems (Figs. 4e and 4f, Supplementary Figs. 17, 18). Impressively, UiO-67-Ir-$Pd(OAc)_2$ exhibits up to 25 times of TONs and 74 times of TOFs over the homogeneous counterparts for the C−H alkenylation reaction of 2-phenylphenol. The superior stability of the MOFs based Pd/photoredox catalyst UiO-67-Ir-$Pd(OAc)_2$ was also confirmed by PXRD, ICP-MS, and XPS (Fig. 2d, Supplementary Figs. 19, 21). Under the optimal reaction conditions, the substrate scope was explored. The results as shown in Table 2 indicated the general applicability of the MOFs based Pd/photoredox catalysis for these oxidation reactions.

These above results clearly show that MOFs based Pd/photoredox catalyst has an overwhelming advantage to address the Pd reoxidation problem in oxidative transformations. The anchor and stabilization of the $Pd^{II}$ catalysts and $Ir^{III}$ PS onto the confined MOFs framework with spatial proximity not only restrain the $Pd^0$ aggregation and facilitates $Pd^0$ reoxidation processes under mild reaction condition, but also realize the reusability of noble metals Pd and Ir, presenting the combined merits of high efficiency, atom economy, sustainability, and environmental-friendliness. Therefore, the MOFs based Pd/photoredox catalysis indeed serves as a promising alternative strategy to achieve efficient Pd catalyst turnover by resolving the fundamental reoxidation problem in Pd-catalyzed oxidation reaction.

**Mechanism investigations**. Thanks to the well-defined dual catalytic sites structure within MOFs framework, an in-depth mechanism investigation for the stepwise electron transfer process and the Pd reoxidation route was conducted. Firstly, we performed a control experiment for MOFs-catalyzed decarboxylative coupling of allylic alcohols. When operated in dark, distinct Pd NPs with a size of approximately 5 nm were observed over the framework of the recovered MOF catalyst, which is termed as UiO-67-Ir-PdNPs (Supplementary Fig. 23c). The Pd NPs nature was confirmed by XPS spectra that the appearance of peaks for Pd $3d_{5/2}$ at 335.1 and Pd $3d_{3/2}$ at 340.5 eV (Supplementary Fig. 23a),

respectively, prove the zero valent Pd, while the $4f_{7/2}$ and $4f_{5/2}$ Ir peaks do not almost change (Supplementary Fig. 23b), indicating that Ir is still +3 oxidation state. In contrast, no detectable $Pd^0$ NPs were found in the recovered UiO-67-Ir-$Pd(TFA)_2$ catalyst under the optimal reaction conditions even after five consecutive runs (Supplementary Fig. 22a). This finding demonstrates the reoxidation of in situ generated $Pd^0$ to $Pd^{II}$ is triggered by visible light illuminate rather than the oxygen oxidation.

Subsequently, we resorted to a robust tool, ultrafast fs-TAS, to track in real-time the stepwise electron transfer. The fs-TAS of the recovered UiO-67-Ir-PdNPs was performed to investigate the charge-transfer dynamics between $Pd^0$ and photoexcited $Ir^{III}$ species. For comparison, the fs-TAS experiment of UiO-67-Ir was also conducted to illustrate the intrinsic excited state dynamics of $Ir^{III}$ species in the absence of Pd-moiety. The pump laser of the fs-TAS experiments was chosen at 400 nm, which could effectively excite the MOFs according to the UV-vis spectra (Supplementary Fig. 10). As shown in Figs. 5a and 5c, UiO-67-Ir and UiO-67-Ir-PdNPs implied similar overall transient absorption profiles in the probing wavelengths ranging from 400-660 nm, comparable to those of other related $Ir^{III}$ complexes[54,55]. Note that the absorption band of UiO-67-Ir-PdNPs shifts to longer wavelength in contrast to UiO-67-Ir, which might be ascribed to charge-transfer interactions between Pd NPs and Ir-moiety within the framework, as observed in other donor-acceptor systems[56,57]. This assignment was further confirmed by the fs-TAS decay kinetics of UiO-67-Ir and UiO-67-Ir-PdNPs. Since the relaxation kinetics largely depends on the probing wavelength, their global fits of the kinetic traces ranging from 450 to 625 nm with 25 nm interval (8 traces) were conducted to evaluate the decay constants. It turned out that three-exponential decay function fits well with the decay kinetic traces of these two MOFs. For UiO-67-Ir, three-exponential components with time constants of 0.39 ps ($\tau_1$), 183 ps ($\tau_2$) and >7 ns ($\tau_3$) were obtained for the description of the transient spectrum (Fig. 5b). We attributed the first component to the dynamics of the $[Ir^{III}]^*$ $^3$MLCT (triplet metal to ligand charge transfer) state through an ultrafast intersystem crossing[58,59]. The second component can be ascribed to the vibrational relaxation to the respective lowest lying vibrational state, followed by some certain charge-transfer processes, which enables the electrons into a shallow trap state[60,61]. The long-living component (>7 ns) might be due to the final relaxation of the $^3$MLCT state. The relaxation kinetics of UiO-67-Ir-PdNPs also suggested three-exponential components with time constants of 0.38 ps ($\tau_1$), 39 ps ($\tau_2$), and 2.6 ns ($\tau_3$) (Fig. 5d). While $\tau_1$ are quite close to these two complexes, the charge-transfer process ($\tau_2$) for UiO-67-Ir-PdNPs is approximate 5 times faster than that of UiO-67-Ir, indicating the opening of another charge-transfer channel due to the existence of $Pd^0$ species. This new channel is due to the electron injection from $Pd^0$ to $[Ir^{III}]^*$, rather than a charge separation state with Pd NPs as the electron mediators, especially considering that no Pd NPs is generated under optimal reaction condition. This result unambiguously suggested the ultrafast electron transfer from $Pd^0$ to $[Ir^{III}]^*$, which facilitates the rapid Pd reoxidation process within the framework.

We further explored the inherent origin of the high efficiency of UiO-67-Ir-Pd$X_2$ over their homogeneous mixed Pd/photoredox system. The photoluminescence (PL) quenching experiments were conducted to evaluate the electron transfer rates of the MOF and homogeneous mixed systems. For the sake of exact comparison with fewer error, we synthesized the Pd NPs supported MOF material UiO-67-PdNPs and poly(N-vinyl-2-pyrrolidone) (PVP) stabilized Pd NPs in which the size of the particles is about 3–5 nm. In homogeneous mixed system, only minimal PL quenching

**Table 2 Substrate scope of representative oxidation reactions catalyzed by UiO-67-Ir-PdX₂ᵃ.**

*Decarboxylative coupling of allylic alcohols:*

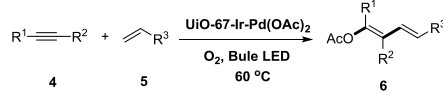

Yield[b]: 91 %, TON: 182

Yield[b]: 93%, TON: 185

Yield[b]: 90 %, TON: 180

Yield[b]: 85%, TON: 169

Yield[b]: 94%, TON: 187

Yield[b]: 81%, TON: 162

*Acetoxypalladation of various alkynes with alkenes:*

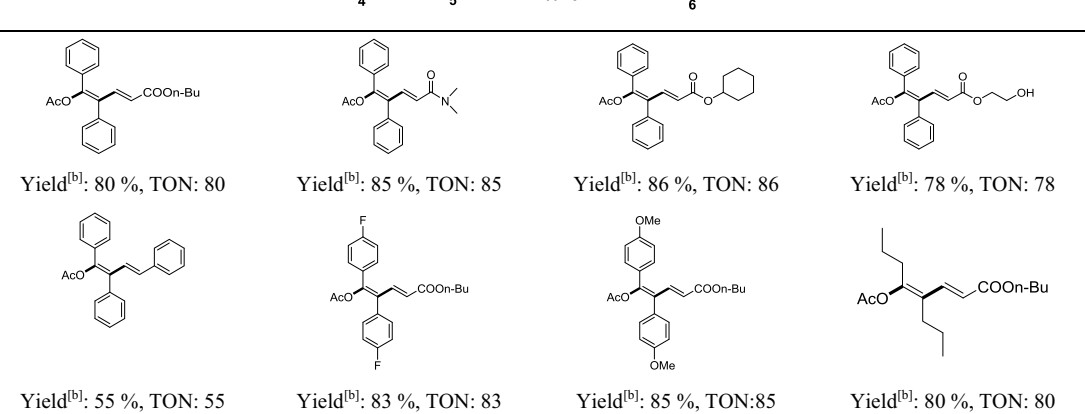

Yield[b]: 80 %, TON: 80

Yield[b]: 85 %, TON: 85

Yield[b]: 86 %, TON: 86

Yield[b]: 78 %, TON: 78

Yield[b]: 55 %, TON: 55

Yield[b]: 83 %, TON: 83

Yield[b]: 85 %, TON:85

Yield[b]: 80 %, TON: 80

*C−H alkenylation of 2-phenylphenol:*

Yield[b]: 77 %, TON: 154

Yield[b]: 80 %, TON: 160

Yield[b]: 78 %, TON: 156

Yield[b]: 75 %, TON: 150

Yield[b]: 60 %, TON: 120

Yield[b]: 64 %, TON: 128

Yield[b]: 50 %, TON: 100

ᵃ0.5 mmol scale.
ᵇIsolated yield.

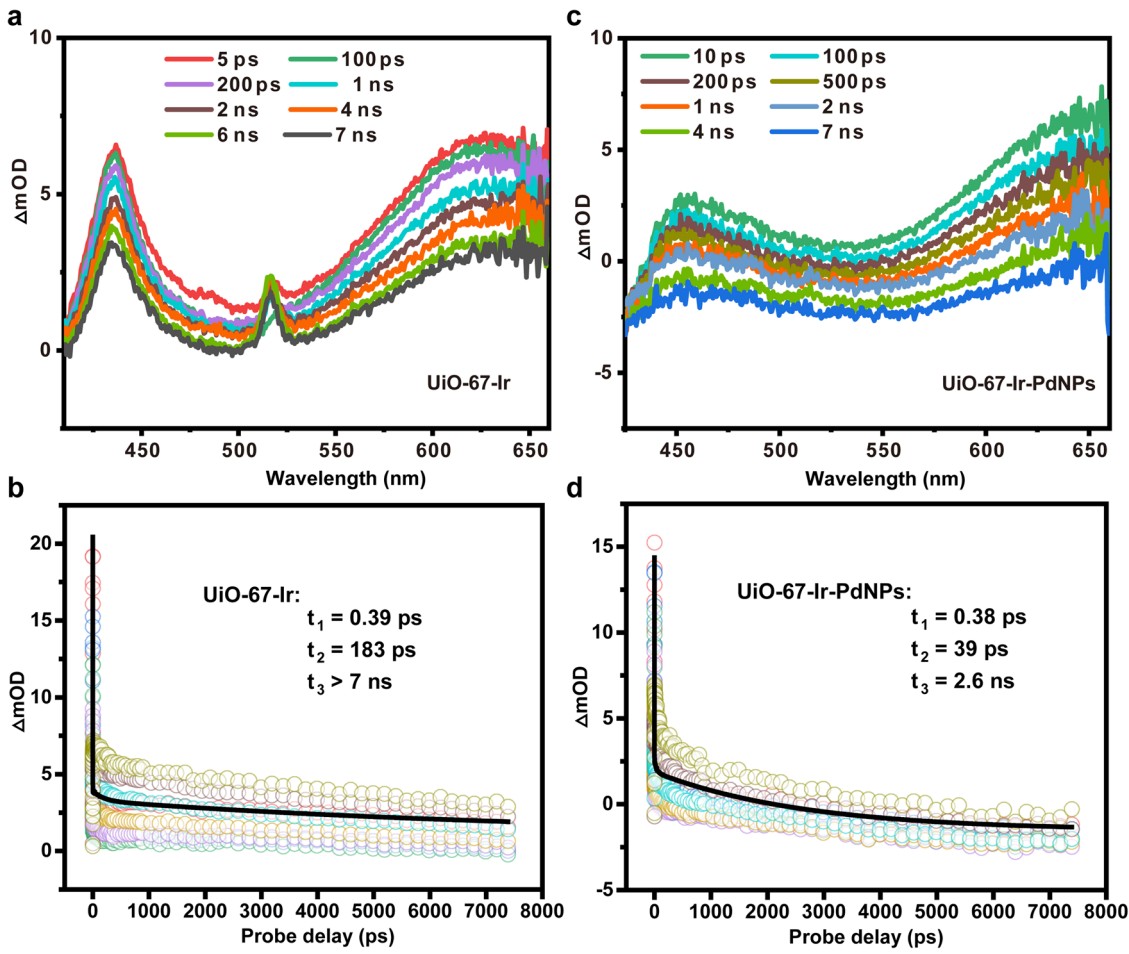

**Fig. 5 fs-TA characteristics of the selected catalysts.** fs-TA spectroscopy of **a** UiO-67-Ir and **c** UiO-67-Ir-PdNPs. fs-TA decay kinetics of **b** UiO-67-Ir and **d** UiO-67-Ir-PdNPs.

occurred when UiO-67-PdNPs or PVP-PdNPs were added into Ir(ppy)$_3$ solution (Figs. 6b and 6c), while a drastic decrease in PL intensity was observed in single MOF system with the increase of Pd NPs loading within UiO-67-Ir-PdNPs (Fig. 6a), illustrating that the close proximity between Ir$^{III}$ PS and Pd NPs is crucial to the electron transfer. Furthermore, These PL quenching curves were fitted with the Stern-Völmer equations to give the quenching constants of $4.85 \pm 0.04$ for UiO-67-Ir-PdNPs and $0.26 \pm 0.02$ or $0.33 \pm 0.01$ for homogeneous controls (Fig. 6d, Supplementary Figs. 26, 27), respectively, which suggested an approximate up to 19 times higher of the electron transfer rates of MOFs over homogeneous mixed systems. This result clearly implied the capability of MOFs based Pd/photoredox catalysis to facilitate the electron transfer between Pd$^0$ species and Ir PS, which contributes to the greatly improved Pd reoxidation efficiency compared to the homogeneous counterpart.

On the basis of these experimental results, we proposed the MOFs based Pd/photoredox-catalyzed oxidation reaction mechanisms (Fig. 7, Supplementary Figs. 29, 30). The stabilized Pd$^{II}$ and Ir$^{III}$ PS catalysts with uniform distribution in MOFs framework work synergistically where the single-site Pd$^{II}$ enables the oxidation process to give the product and release Pd$^0$ species, the latter could be in situ oxidized to active Pd$^{II}$ by the excited [Ir$^{III}$]* to complete the rapid Pd catalyst turnover. We consider that the cycles of Pd species in these oxidation reactions are the same as the previously reported Pd/stoichiometric-oxidant systems except for the Pd catalyst turnover pathway. Taking Pd-catalyzed C−H alkenylation of 2-phenylphenol as an

example, the reaction was initiated by hydroxyl-directed insertion of Pd$^{II}$ into the C−H bond of **7** to form cyclopalladated intermediate. A subsequent coordination and then Heck cross-coupling of the cyclopalladated intermediate with alkenes, followed by further β-H elimination gave the targeted product and release the Pd$^0$ species. Owing to the stabilization of the Pd$^0$ species and the close proximity of Pd and Ir sites within the framework, a rapid SET process would occur between the in situ generated Pd$^0$ species and [Ir$^{III}$]*, leading to the fast regeneration of active Pd$^{II}$ catalyst and Ir$^{II}$ species. At last, the Ir$^{II}$ species is reoxidized to Ir$^{III}$ by O$_2$ trapped in the MOF framework to finish the photocatalytic cycle. It is worth noting that MOF can trap O$_2$ in the pores, and thereby create a local high-pressure O$_2$ atmosphere, thus further facilitating the oxidation of Ir$^{II}$ species and simultaneously producing superoxide anion radical (O$_2$$^{-\bullet}$) that can be detected by electron paramagnetic-resonance spectroscopy experiments (Supplementary Fig. 28).

## Discussion

In summary, we reported a promising alternative strategy to resolve the fundamental Pd reoxidation problem in Pd-catalyzed oxidation reactions by incorporating Ir$^{III}$ PS and Pd$^{II}$ catalysts into MOFs framework. The capability of the resulted UiO-67-Ir-Pd$X_2$ to restrain the Pd$^0$ aggregation and accelerate electron transfer between Pd$^0$ species and photoexcited Ir$^{III}$ PS facilitates the ultrafast Pd catalyst turnover, which make them outstanding candidates in catalyzing three representative Pd-catalyzed oxidation reactions with low Pd consumptions, atom-economy, recyclability and

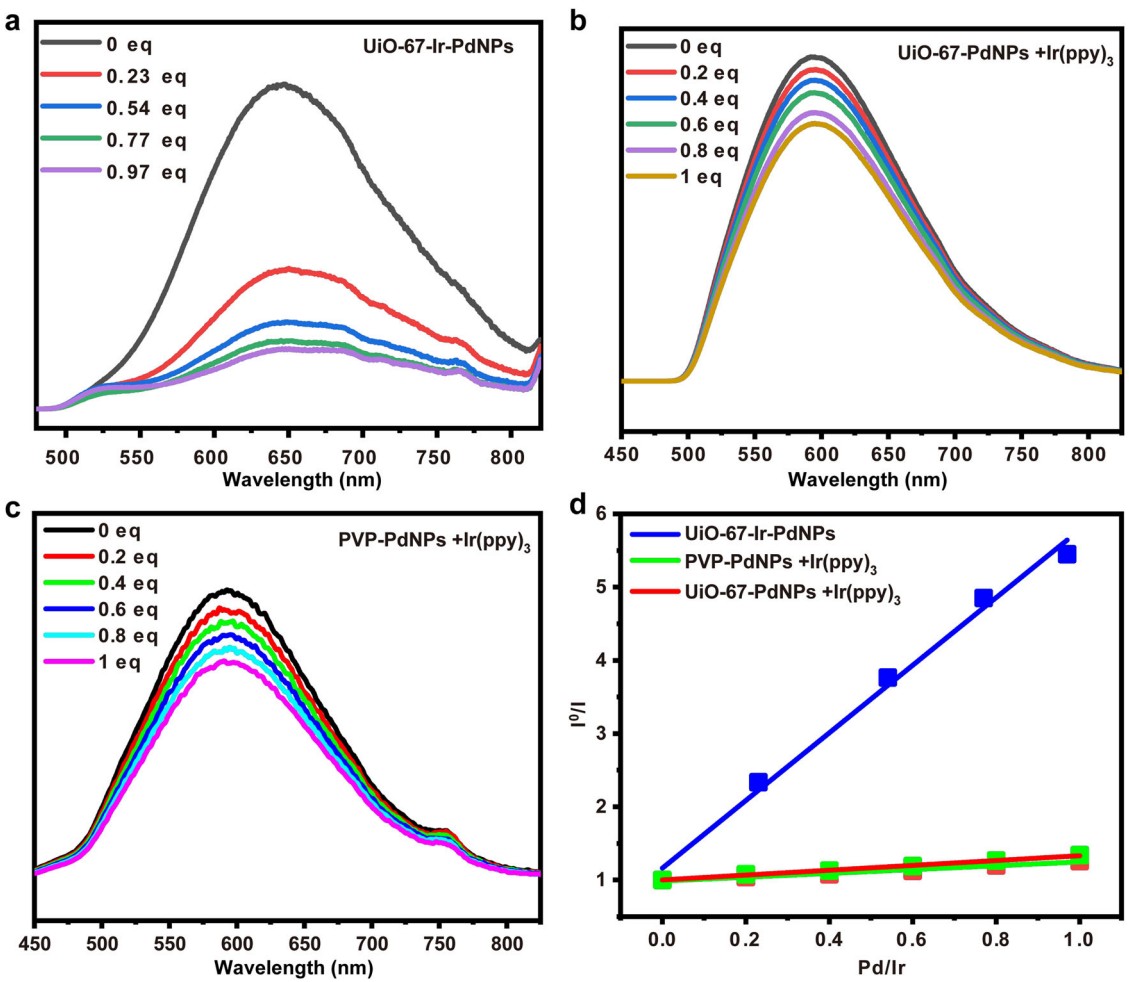

**Fig. 6 PL measurements of the selected catalysts.** PL quenching curves of (**a**) UiO-67-Ir-PdNPs with different Pd NP loadings (**b**) Ir(ppy)$_3$ with different loadings of UiO-67-PdNPs. (**c**) Ir(ppy)$_3$ with different loadings of PVP-PdNPs. (**d**) Plots of I$^0$/I as a function of the ratio of Pd NPs to Ir$^{III}$ PS.

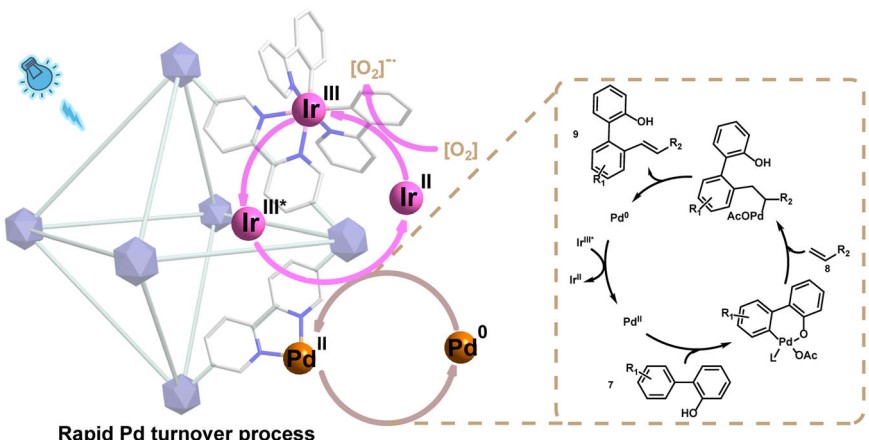

**Fig. 7 Proposed mechanism for MOFs based Pd/photoredox catalysis.** Taking the Pd-catalyzed C−H alkenylation of 2-phenylphenol as an example.

environmental-friendliness. Intensive mechanism investigations demonstrated the successful application of MOFs based Pd/photoredox catalysis in Pd-catalyzed oxidation reactions by regulating the Pd$^0$ aggregation and reoxidation processes. This work suggested the potential of MOFs based Pd/photoredox catalysis as a general strategy for other valuable Pd-catalyzed oxidative transformations by incorporating diverse Pd and photocatalysts into MOFs backbone. We also envision this strategy can be applied in broader

transition metal (Ru, Rh, etc.) catalyzed oxidative transformations where the catalyst turnover by reoxidation of a reduced transition metal is the rate-limiting process.

## Methods

**Synthesis of UiO-67-Ir.** UiO-67-Ir was synthesized through solvothermal reaction of ZrCl$_4$ (0.24 mmol, 55.5 mg), **H$_2$L** (0.02 mmol, 15 mg), bpy (0.18 mmol, 45 mg) and acetic acid (10.8 mmol, 618 μL) in a solvent of DMF (7.5 mL) at 120 °C for

24 h. Then, the reaction was cooled to room temperature at a rate of 20 °C/h. The product was centrifuged and washed with DMF and acetone three times respectively. After that, the solid was immersed in fresh acetone for three days to exchange the high boiling-point DMF. The acetone was refreshed every other day during the period. Finally, the solid was heated at 50 °C under vacuum to remove the trapped solvents in MOFs pore.

**Synthesis of UiO-67-Ir-PdX$_2$ (X = OAc, TFA).** UiO-67-Ir (50 mg, 0.014 mmol based on Ir) was placed in 3 mL acetone of Pd(TFA)$_2$ (4.6 mg, 0.014 mmol) or Pd(OAc)$_2$ (3 mg, 0.014 mmol) solutions. The mixtures were sonicated for 10 min and then stirred at 50 °C for 24 h. After that, the mixtures were centrifuged and the solids were washed with acetone three times. The solids were immersed in acetone for 3 days, and acetone was refreshed every other day during the period. After soaking, the solids were collected via centrifugation and dried at 50 °C under vacuum to obtain UiO-67-Ir-PdX$_2$ (X = OAc, TFA).

**Synthesis of UiO-67-Ir-PdNPs with different Pd NPs loadings.** To a 4 mL glass vial was added 1 mL acetone suspension of UiO-67-Ir (6.0 μmol based on Ir) and Pd(TFA)$_2$ of different amounts (0.25, 0.5, 0.75, and 1.0 eq of Ir). The reaction mixture was stirred at 50 °C for 24 h. The solid was collected by centrifugation and washed with acetone three times and then immersed in fresh acetone for 3 days. The acetone was refreshed every other day during the period. Subsequently, the solid was heated at 50 °C under vacuum to remove the trapped solvents in the pores. Finally, the samples of UiO-67-Ir-Pd(TFA)$_2$ with different Pd/Ir ratios were added in Pd-catalyzed decarboxylative coupling of allylic alcohols under the optimal reaction condition but without light to obtain UiO-67-Ir-PdNPs. The ratio between Pd and Ir for these recovered UiO-67-Ir-PdNPs was determined to be 0.23, 0.54, 0.77, 0.97, respectively.

**Synthesis of UiO-67-PdNPs.** ZrCl$_4$ (70.0 mg), [(5,5′-dicarboxy-2,2′-bipyridine-) palladium(II)] dichloride (4.2 mg) and H$_2$bpdc (H$_2$bpdc, para-biphenyldicarboxylic acid) (70.24 mg) were dispersed in DMF (10 mL), sealed in a 20 mL vial at 100 °C for 36 h. Then, the reaction was cooled to room temperature at a rate of 20 °C/h. The produced powders were isolated by centrifugation and dried at ambient temperature. Subsequently, the as-synthesized sample was soaked in chloroform for three 18 h periods at room temperature to remove DMF and ligand precursors, then filtered off and dried under vacuum at room temperature for 24 h. Finally, the as-synthesized sample was treated in the stream of H$_2$ at 250 °C for 4 h to obtain the UiO-67-PdNPs. The Pd NP loadings in UiO-67-PdNPs were determined to be 1.0 wt%.

**Synthesis of poly(N-vinyl-2-pyrrolidone) (PVP) stabilized Pd NPs.** In a round bottle, aqueous Na$_2$PdCl$_4$ solution (1.0 × 10$^{-4}$ mol·cm$^{-3}$) was added to a preheated (90 °C) aqueous mixture (30 cm$^3$) containing ascorbic acid (8.5 × 10$^{-4}$ mol) and PVP (5.0 × 10$^{-3}$ mol; PVP/Pd molar ratio = 10). The mixture was kept under stirring at 90 °C for 3 h, and a solution of Pd NPs was obtained. The Pd NPs were cleaned up from the excess of PVP via flocculation with acetone (1/3 v/v solution/acetone), rinsed thoroughly with acetone, and redispersed in water attaining ca. 0.4 wt% Pd in the final solution.

**General procedure for the MOFs based Pd/photoredox-catalyzed decarboxylative coupling of allylic alcohols.** A mixture of **1** (0.5 mmol), UiO-67-Ir-Pd(TFA)$_2$ (0.5 mol% based on Pd) was added in a 25 mL schlenk tube with a magnetic stir bar. The tube was outgassed completely and purged with O$_2$ for three cycles. Then, the toluene (2.85 mL) and DMSO (0.15 mL) solution of **2** (0.6 mmol) was injected into the mixture and the tube was charged with a O$_2$ balloon. Subsequently, the tube was placed into a constant temperature incubator to reduce the impact of the photo-induced heat. The mixture was vigorously stirred under a 40 W blue LED irradiation at room temperature for 18 h. After completion of the reaction, the mixture was centrifuged to remove the solid phase, and the filtrate was extracted with ethyl acetate (3 × 10 mL). The combined ethyl acetate layer was then dried over sodium sulfate and concentrated under vacuum. The resulting crude product was purified by silica gel chromatography to afford the desired product.

**General procedure for the MOFs based Pd/photoredox-catalyzed acetoxypalladation of various alkynes with alkenes.** A mixture of **4** (0.5 mmol), UiO-67-Ir-Pd(OAc)$_2$ (1 mol% based on Pd), and KBr (40 mol%, 23.8 mg) was added in a 25 mL schlenk tube with a magnetic stir bar. The tube was outgassed completely and purged with O$_2$ for three cycles. Then, the MeCN (2 mL) and HOAc (0.5 mL) solution of **5** (1 mmol) was injected into the mixture and the tube was charged with a O$_2$ balloon. Subsequently, the tube was placed into a constant temperature incubator to reduce the impact of the photo-induced heat. The mixture was vigorously stirred under a 40 W blue LED irradiation at 60 °C for 24 h. After completion of the reaction, the mixture was centrifuged to remove the solid phase. The filtrate was poured into the saturated NaHCO$_3$ solution and then extracted with ethyl acetate (3 × 10 mL). The combined ethyl acetate layer was then dried

over sodium sulfate and concentrated under vacuum. The resulting crude product was purified by silica gel chromatography to afford the desired product.

**General procedure for the MOFs based Pd/photoredox-catalyzed C−H alkenylation of 2-phenylphenol.** A mixture of **7** (0.5 mmol) and UiO-67-Ir-Pd(OAc)$_2$ (0.5 mol% based on Pd) was added in a 25 mL schlenk tube with a magnetic stir bar. The tube was outgassed completely and purged with O$_2$ for three cycles. Then, the DMSO (2 mL) solution of **8** (1 mmol) was injected into the mixture and the tube was charged with a O$_2$ balloon. Subsequently, the tube was placed into a constant temperature incubator to reduce the impact of the photo-induced heat. The mixture was vigorously stirred under a 40 W blue LED irradiation at 50 °C for 12 h. After completion of the reaction, the mixture was centrifuged to remove the solid phase. The filtrate was poured into the aqueous solution and then extracted with ethyl acetate (3 × 10 mL). The combined ethyl acetate layer was then dried over sodium sulfate and concentrated under vacuum. The resulting crude product was purified by silica gel chromatography to afford the desired product.

**Characterizations.** Powder X-ray diffraction (PXRD) patterns were collected on a Bruker D8 powder diffractometer at 40 kV, 40 mA with Cu Kα radiation (λ = 1.5406 Å), with a step size of 0.01995° (2θ). Thermogravimetric analyses (TGA) were performed on a Q600 SDT instrument under a flow of air at a heating rate of 5 °C/min from 25–900 °C. $^1$H NMR and $^{13}$C NMR were done on a Bruker Model AM-400 (400 MHz) spectrometer. The UV–Vis diffuse reflectance spectra were collected from 200 to 800 nm using PE lambda 750. The content of metal ions was determined by the inductively coupled plasma mass spectrometry (ICP-MS) (Agilent 720ES). The N$_2$ adsorption measurements were performed on a Micro-Active ASAP 2460 systems under 77 K. The X-ray photoelectron spectroscopy (XPS) experiments were conducted using Thermo fisher Scientific with an Al-K-Alpha+ radiation source. Scanning-electron microscopy (SEM) was recorded on MERLIN Compact. Transmission-electron microscopy (TEM) and elemental mapping images were performed on FEI Tecnai G2 F20. The steady-state PL measurements were carried out by a fluorescence spectrophotometer with an excitation wavelength of 400 nm. The electron paramagnetic-resonance (EPR) spectroscopy was recorded on a BRUKER ELEXSYS-II E500 CW-EPR electron paramagnetic-resonance spectrometer under dark or blue LED irradiation.

The Pd K-edge X-ray absorption fine structure (XAFS) measurements were performed at BL14W1 beamline in Shanghai Synchrotron Radiation Facility (SSRF), China. The hard X-ray was monochromatized with Si (311) double-crystal monochromator and the XAFS data were collected in transmission mode in the energy range from −200 below to 1000 eV above the Pd K-edge. The XAFS spectra of [Ir(bpy)(ppy)$_2$](PF$_6$) and UiO-67-Ir-PdX$_2$ (X = OAc, TFA) at Ir L$_3$-edge were collected on the beamline BL07A1 in National Synchrotron Radiation Research Center (NSRRC). The radiation was monochromatized by a Si (111) double-crystal monochromator.

The fs-TA experiments were performed on a Helios pump-probe system (Ultrafast Systems LLC) coupled with an amplified femtosecond laser system (Coherent, 35 fs, 1 kHz, 800 nm). A small portion (around 10 μJ) of the fundamental 800-nm laser pulses was focused into a 1-mm CaF$_2$ to generate the probe pulses (from 400 to 650 nm). The 400-nm pump pulses were generated from an optical parametric amplifier (TOPAS-800-fs).

## Data availability

The data supporting the findings of this study are available within the article and its Supplementary Information files. Any other data that support the findings of this study are available from the corresponding author upon reasonable request. Source data are provided with this paper.

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

## Acknowledgements

H.F.J. acknowledges support from the Key-Area Research and Development Program of Guangdong Province (2020B010188001). J. W. L. acknowledges supports from the Guangdong Basic and Applied Basic Research Foundation (2020A1515110247), the China Postdoctoral Science Foundation (2019M660199), and from the Fundamental Research Funds for the Central Universities (2019MS043). Y.W.R. acknowledges supports from the Guangdong Basic and Applied Basic Research Foundation (2021A1515010076).

## Author contributions

J.L. conducted the experiments, analyzed the results, and wrote the manuscript. Y.R. and H.J. designed the research, supervised the project, and edited the manuscript. L.H. participated in writing the manuscript. Q.L. performed some experimental data analysis.

## Competing interests

The authors declare no competing interests.
