## [Peer Review File · Nature Communications]

Title: Visible Light-Driven Efficient Palladium Catalyst Turnover in Oxidative Transformations within Confined FrameworksREVIEWER COMMENTS

Reviewer #1 (Remarks to the Author):

The authors incorporated Ir and Pd complexes into the matrix of UiO-67 MOF to obtain the UiO-67-Ir-PdX₂ (X = OAc, TFA). The UiO-67-Ir-PdX₂ (X = OAc, TFA) shows impressive catalytic properties towards a couple of oxidation reactions. The catalytic mechanism is discussed in detail. However, I have several serious concerns in the structural characterization of UiO-67-Ir-PdX₂ (X = OAc, TFA) before and after characterization. I do not recommend its publication until the authors fully address the following comments.

1. The authors characterized the structure of obtained complexes by XRD, EDX maps and XAFS. However, from the presented analyses, none of them could be evident for the structure showing in Figure 1a. Please note that EDX maps only showed the elemental distribution rather than the crystal structure. Strong evidence on the crystal structure must be required. A combination of the fitting process of the XRD patterns, elemental analyses, IR and Raman spectra must be helpful.
2. The authors also showed the UiO-67-Pd in Table 1, what is the structure?
3. For the XPS and XAFS measurements, the authors should provide the information of pure Ir and Pd complex for comparison.
4. For the XAFS results,
 - a. Please make sure that the Ir spectra are tested in the K-edge or L3-edge. Ir-K has quite high energy.
 - b. Please also provide the original XAFS data in E- and k-space in addition to the R-space.
 - c. Every path showing in Fig.2b should be explained in detail.
5. In the catalytic reaction, the catalytic performance of the physical mixture of these three complexes is strongly recommended to be tested as a control group.
6. In Figure S14, the XRD pattern is required to a more extended range to show the feature of Pd NPs.
7. In Figure S17, how the authors confirmed it is Pd NPs rather than Ir NPs or others? Other characterizations are strongly required to confirm this point. Also, the morphology of UiO-67-Ir-Pd(TFA)₂ showing in Figure S17a is different from the original morphology as well as the morphology showing in Figure S17b. This suggests some structure changes. Please identify and explain these differences.

Reviewer #2 (Remarks to the Author):

REF: H.Jiang_PhotocatMOF_NatCommun_317681_0

Title: Visible Light-Driven Efficient Palladium Catalyst Turnover in Oxidative Transformations within Confined Frameworks

This research paper by Ren, Jiang, and co-workers focuses on immobilizing molecular Pd-catalysts and Ir-photosensitizers within the MOF UiO-67 for atom-efficient photocatalytic oxidative transformations.

This referee found the paper to be interesting with a thorough material characterization and catalytic rationale. The scope of analysis methods and substrates tested for oxidative transformations support

the claims well and are coherent. The topic of Pd oxidations circumventing harsh conditions and stoichiometric additives is fascinating and is advanced by this work showing improvements over homogeneous conditions.

Having said this, the general PS-catalyst electro-communication in UiO-67 approach described in this manuscript as well as the incorporation of molecular Pd catalysts in UiO-67 are not new and have been published several times over the last ~6 years. Comparable studies using the same conceptual approach with UiO-67 (molecular catalyst and photosensitizer) with similar conclusions that this reviewer is aware of: e.g., *J. Phys. Chem. C* 2018, 122, 3305, DOI: 10.1021/acs.jpcc.8b00471; *J. Am. Chem. Soc.* 2016, 138, 8698, DOI: 10.1021/jacs.6b04552; *Catal. Lett.* 2021, DOI: 10.1007/s10562-021-03719-0, *J. Am. Chem. Soc.* 2014, 136, 6566, DOI: 10.1021/ja5018267. This questions whether the strategy originality enables publication in *Nature Communications*. In addition, although interesting, the time-resolved spectroscopic experiments have been conducted in presence of Pd nanoparticles instead of the molecular complexes (in UiO-67-Ir-Pd) which limits the transferability of the conclusions to the actual case studied. If the authors can provide a more convincing argumentation and evidence of a conceptual novelty and a deeper fundamental understanding of mechanisms pointing towards an improved rational design of such systems, the work may well become suitable for *Nature Communications*.

Independently, this referee suggests considering the following points:

1. Scheme 1 is text-heavy and could be more suited as an “abstract/TOC” image. Particularly panel B is pure text and the Scheme may benefit from enlarging the benefits of this work (panel C) in direct contrast to homogeneous conditions (panel A).
2. Statement on line 46 regarding Pd catalyst turnovers often being limited by reoxidation of the Pd0 is lacking supporting references.
3. Related to the comment above on originality, it is warranted to include a short discussion and overview on comparable UiO-67 systems with dual-anchored molecular complexes for photocatalytic applications towards the end of the introduction (line 86).
4. The claim in lines 147-149 relating to the ideal diameter(s) for photocatalytic applications is interesting and is likely correct. However, several literature works should be cited to back up the individual claims of i) efficient sensitization, ii) exposure of more active sites, and iii) effective O₂ adsorption.
5. The EXAFS fit in Figure 2b deviates to the experimental data in several places. A short explanation/clarification would be welcome.
6. The described homogeneous catalytic rationale in lines 212-229 is coherent but lacks citations.
7. In lines 292-294 the post-catalysis crystallinity is discussed and while Figure S14 shows clear retention of reflexes (in-line with the authors’ analysis), a short comment/explanation on the distinct change in relative reflex intensities is warranted.
8. Table 2 is very interesting and shows a nice scope of tested substrates. For each reaction respectively, similar TONs/Yields are obtained. Can the main source of deactivation be distilled from this? Selectivity issues?
9. As stated above, the fs-TAS studies are a welcome addition to the mechanistic analysis. However, from the manuscript it only becomes clear at the end why this was performed on a “control” sample batch specifically including Pd nanoparticles instead of the sample UiO-67-Ir-Pd() actually used in

catalysis. Including the data for the latter would be interesting to attain a full overview on photophysical behaviour instead of focusing on Ir-PdO interactions.

10. The same rationale is frequently repeated in a very similar fashion, first the introduction revolves around these points, afterwards in line 212-229 and 316-336 similar formulations for the same arguments are used.

11. Figure 1 (b,c,d) quality is poor (at least on this referee's computer) which makes its readability limited.

12. This reviewer believes that "superoxygen" | 431 and 433 should be corrected to "superoxide".

13. Sentence 433 should be rephrased since superoxide is unlikely to act as an oxidant.

14. There is likely to be a structure error in fig 3d

Dear Editor and Reviewers:

Thank you very much for your letter and the reviewer's comments concerning our manuscript entitled "Visible Light-Driven Efficient Palladium Catalyst Turnover in Oxidative Transformations within Confined Frameworks". These comments and suggestions are valuable and helpful for improving our paper. We have made careful corrections according to the reviewer's advices and the changes are marked with a yellow background in the revised manuscript. The responses to the reviewers are as follows:

Reviewer #1 (Remarks to the Author):

The authors incorporated Ir and Pd complexes into the matrix of UiO-67 MOF to obtain the UiO-67-Ir-PdX₂ (X = OAc, TFA). The UiO-67-Ir-PdX₂ (X = OAc, TFA) shows impressive catalytic properties towards a couple of oxidation reactions. The catalytic mechanism is discussed in detail. However, I have several serious concerns in the structural characterization of UiO-67-Ir-PdX₂ (X = OAc, TFA) before and after characterization. I do not recommend its publication until the authors fully address the following comments.

Response: We thank this reviewer for the positive comments.

1. The authors characterized the structure of obtained complexes by XRD, EDX maps and XAFS. However, from the presented analyses, none of them could be evident for the structure showing in Figure 1a. Please note that EDX maps only showed the elemental distribution rather than the crystal structure. Strong evidence on the crystal structure must be required. A combination of the fitting process of the XRD patterns, elemental analyses, IR and Raman spectra must be helpful.

Response: As the reviewer said, the characterizations of XRD, EDX mapping and XAFS could not be evident for the exact crystal structure of the two catalysts, a combination of multiple characterization techniques is helpful to clarify the

microstructure of the catalysts. According to the reviewer's advices, a combination of various characterization techniques (XRD, XPS, ^1H NMR, TGA, ICP-MS, UV/vis, IR and EXAFS) were used in the revised manuscript to corroborate the microstructure of the catalysts. Some supplementary characterizations, such as IR spectra of catalysts, XPS and EXAFS spectra for pure Ir and Pd complexes (Question 3 proposed by the reviewer) were added in the revised manuscript. In view of these characterizations, the discussion of the characterization of catalyst structure has been modified, which can evidence the crystal structure showing in Fig. 1a (Fig. 2a in the revised manuscript).

In brief, the purpose of the test characterization technology used in this manuscript is as follows. Firstly, the almost same XRD patterns of UiO-67-Ir-PdX₂ (X = OAc, TFA) and UiO-67 prove the same topology structure of them. The crystal structure of UiO-67 has been previously reported (*J. Am. Chem. Soc.* **2008**, *130*, 13850-13851), which is composed of 12-connected [Zr₆O₄(OH)₄]¹²⁺ clusters and biphenyl-4,4'-dicarboxylate (BPDC) ligands. With the topological framework of UiO-67 in hand, ^1H NMR, TGA and ICP-MS as elemental analyses characterizations were used to determine the ratio of different ligands and the contents of metal ion (Zr⁴⁺, Ir³⁺, Pd²⁺). The existence of important functional groups in UiO-67-Ir-PdX₂ (X = OAc, TFA) was judged qualitatively by UV/vis and IR spectra. XPS and EXAFS spectra were used to clarify the valence and coordination environments of Ir and Pd in the catalysts. Thus, the clear structures of obtained catalysts showing in Fig. 1a (in the original manuscript) can be confirmed by a combination of multiple characterizations, although we could not get the crystal structure of the catalysts.

2. The authors also showed the UiO-67-Pd in Table 1, what is the structure?

Response: UiO-67-Pd was synthesized according to previous reports (*Green Chem.* **2014**, *16*, 3978; *ACS Catal.* **2016**, *6*, 6324). The synthesis procedure and structure were placed in Section 2.2.4 and Supplementary Fig. 6 in the Supplementary Information.

3. For the XPS and XAFS measurements, the authors should provide the information of pure Ir and Pd complex for comparison.

Response: According to the reviewer's suggestion, the XPS and XAFS measurements of pure Ir and Pd complexes have been conducted for comparison. The corresponding results have been included in the revised manuscript (Fig. 3) and supplementary information (Supplementary Figs. 12-14 and Fig. 16).

4. For the XAFS results,

a. Please make sure that the Ir spectra are tested in the K-edge or L₃-edge. Ir-K has quite high energy.

Response: The XAFS of Ir was tested in L₃-edge, not K-edge. We are regretful for the typing error in the original manuscript and we appreciate the careful examination of this reviewer.

b. Please also provide the original XAFS data in E- and k-space in addition to the R-space.

Response: According to the reviewer's suggestion, the XAFS data in E- and k-space have been included in the revised manuscript (Fig. 3) and supplementary information (Supplementary Fig. 15).

c. Every path showing in Fig.2b should be explained in detail.

Response: In order to interpret each path showing in Fig. 2b (in original manuscript) more clearly, we provided the molecular model of the Ir^{III} complex within UiO-67-Ir-PdX₂ (X = OAc, TFA) in the revised supplementary information, and the atoms of each path for corresponding Ir-C/N bonds have been labeled (Supplementary Fig. 16). Relative description has been added in the revised manuscript.

5. In the catalytic reaction, the catalytic performance of the physical mixture of these three complexes is strongly recommended to be tested as a control group.

Response: According to the reviewer's suggestion, the catalytic performance of the physical mixture of three complexes have been tested and the corresponding catalytic results have been added in Table 1 (entry 12), Supplementary Table 6 (entry 12) and Table 7 (entry 12).

6. In Figure S14, the XRD pattern is required to a more extended range to show the feature of Pd NPs.

Response: The PXRD patterns in Figure S14 (in original Supporting Information) have been extended to 2θ 50, and this Figure was renamed as Supplementary Fig. 19 in the revised Supplementary Information. This Figure shows no distinct characteristic peaks of Pd NPs were observed in the recovered catalysts UiO-67-Ir-PdX₂ (X = OAc, TFA) after catalysis. This result proved again that no obvious Pd NPs were formed under the optimal reaction conditions.

7. In Figure S17, how the authors confirmed it is Pd NPs rather than Ir NPs or others? Other characterizations are strongly required to confirm this point. Also, the morphology of UiO-67-Ir-Pd(TFA)₂ showing in Figure S17a is different from the original morphology as well as the morphology showing in Figure S17b. This suggests some structure changes. Please identify and explain these differences.

Response: We firstly appreciate the reviewer's suggestion for the confirmation of Pd NPs. The XPS measurements of the recovered catalysts operated in dark were conducted (Supplementary Figs. 23a and 23b). The result showed that no characteristic peaks of Ir⁰ was found and the oxidation state of Ir is still +3. While in Pd region, two peaks at 335.1 and at 340.5 eV, attributing to Pd⁰ 3d_{5/2} and Pd⁰ 3d_{3/2}, respectively, were observed. These results undoubtedly show that the formed nanoparticles should be Pd NPs. Relative description and discussion have been added in the revised manuscript, as follows:

When operated in dark, distinct Pd NPs with a size of approximate 5 nm were

observed over the framework of the recovered MOF catalyst, which is termed as UiO-67-Ir-PdNPs (Supplementary Fig. 23c). The Pd NPs nature was confirmed by XPS spectra that the appearance of peaks for Pd 3d_{5/2} at 335.1 and Pd 3d_{3/2} at 340.5 eV (Supplementary Fig. 23a), respectively, prove the zero valent Pd, while the 4f_{7/2} and 4f_{5/2} Ir peaks do not almost change (Supplementary Fig. 23b), indicating that Ir is still + 3 oxidation state.

Secondly, as for the morphological differences of the recovered catalyst and the fresh catalyst UiO-67-Ir-Pd(TFA)₂, we think it is partly due to the slight structure distortion on the catalyst surface during the catalytic reaction. After all, this TEM picture (Figure S17a which has become to Supplementary Fig. 22a in the revised SI) was taken after the catalyst has been used for five consecutive runs. Nevertheless, the overall morphology of the recovered catalyst still exists as octahedral particles, and their unchanged PXRD patterns (Supplementary Fig. 19) suggested they are still long range ordered without obvious structure destruction. This slight morphology differences before and after catalysis in UiO series MOFs is not uncommon and can also be observed elsewhere (*Chem. Commun.* **2014**, 50, 4810; *J. Am. Chem. Soc.* **2020**, *142*, 20, 9428).

Reviewer #2 (Remarks to the Author):

REF: H.Jiang_PhotocatMOF_NatCommun_317681_0

Title: Visible Light-Driven Efficient Palladium Catalyst Turnover in Oxidative Transformations within Confined Frameworks

This research paper by Ren, Jiang, and co-workers focuses on immobilizing molecular Pd-catalysts and Ir-photosensitizers within the MOF UiO-67 for atom-efficient photocatalytic oxidative transformations. This referee found the paper to be interesting with a thorough material characterization and catalytic rationale. The scope of analysis methods and substrates tested for oxidative transformations support the claims well and are coherent. The topic of Pd oxidations circumventing harsh conditions and stoichiometric additives is fascinating and is advanced by this work

showing improvements over homogeneous conditions.

Having said this, the general PS-catalyst electro-communication in UiO-67 approach described in this manuscript as well as the incorporation of molecular Pd catalysts in UiO-67 are not new and have been published several times over the last ~6 years. Comparable studies using the same conceptual approach with UiO-67 (molecular catalyst and photosensitizer) with similar conclusions that this reviewer is aware of: e.g., J. Phys. Chem. C 2018, 122, 3305, DOI: 10.1021/acs.jpcc.8b00471; J. Am. Chem. Soc. 2016, 138, 8698, DOI: 10.1021/jacs.6b04552); Catal. Lett. 2021, DOI: 10.1007/s10562-021-03719-0, J. Am. Chem. Soc. 2014, 136, 6566, DOI: 10.1021/ja5018267. This questions whether the strategy originality enables publication in Nature Communications. In addition, although interesting, the time-resolved spectroscopic experiments have been conducted in presence of Pd nanoparticles instead of the molecular complexes (in UiO-67-Ir-Pd) which limits the transferability of the conclusions to the actual case studied. If the authors can provide a more convincing argumentation and evidence of a conceptual novelty and a deeper fundamental understanding of mechanisms pointing towards an improved rational design of such systems, the work may well become suitable for Nature Communications.

Response: We firstly appreciate the positive comments from this referee.

As the reviewer said, the merger of transition metal catalysts and photosensitizers into MOFs have been reported in recent years. Different groups have made important progress in this field and found successful applications of these MOF catalysts in organic transformations, water splitting, CO₂ reduction, and etc. We have also stated such research process and cited related literatures in the original manuscript (line 76-80). However, the purpose of this manuscript is to use the designability of MOFs framework to solve the problems existing in Pd-catalyzed oxidation reactions, and use the easy characterization of crystalline MOFs framework to deeply explore the catalytic reaction mechanism, so as to provide a theoretical basis for the design of efficient and green Pd oxidation catalysts.

As reported, Pd-catalyzed oxidation reactions have been emerged as one class of the most powerful, valuable and widely studied methodology in modern organic synthesis (*Chem. Rev.* **2018**, *118*, 2636), and our group have also made much efforts in this field (*J. Am. Chem. Soc.* **2008**, *130*, 5030; *J. Am. Chem. Soc.* **2009**, *131*, 3846; *Angew. Chem. Int. Ed.* **2012**, *51*, 5696; *Acc. Chem. Res.* **2012**, *45*, 1736; *Angew. Chem. Int. Ed.* **2012**, *51*, 7292; *J. Am. Chem. Soc.* **2013**, *135*, 5286; *Angew. Chem. Int. Ed.* **2017**, *56*, 15926; *Angew. Chem. Int. Ed.* **2019**, *58*, 17148). However, the state-of-the-art approaches for this process still significantly deviates from the perspective of high efficiency, atom economy, sustainability and environmental-friendliness. The present work focuses on addressing above issues by applying the MOFs based Pd/photoredox catalysis as a promising new strategy, and we utilized three representative Pd-catalyzed oxidation reactions to verify the rationality of the design of these MOFs based Pd/photoredox catalysts. Therefore, the research idea of this manuscript is to start from the basic problems of Pd-catalyzed oxidation reactions and put forward a new strategy to solve them through reasonable MOFs heterogeneous catalyst design. We also envision this strategy can be applied in broader transition metal (Ru, Rh, etc.)-catalytic oxidative transformations where the catalyst turnover by reoxidation of a reduced transition metal is the rate-limiting process. Finally, according to this reviewer's advices, we revised the Introduction section and the discussion of mechanism, and we hope that the highlight of this work has been clearly presented to the reviewer and reader.

On the other hand, the main reasons why we conducted the time-resolved fs-TA spectroscopic experiments for UiO-67-Ir-PdNPs, instead of UiO-67-Ir-PdX₂, are as following: These three Pd-catalyzed oxidation reactions are initiated by the interaction between Pd^{II} and substrate, accompanied by the generation of Pd⁰ species. The excited [Ir^{III}]* moiety within the framework acts as an oxidant to reoxidize the *in situ* formed Pd⁰ species to Pd^{II}, that is, the electron injection from Pd⁰ to [Ir^{III}]*, which can be monitored by the time-resolved fs-TA spectra and decay constants. While the fs-TA spectroscopic experiment of UiO-67-Ir-PdX₂ cannot reflect the real electron transfer channel and Pd⁰ reoxidation pathway.

In addition, according to the comments of the referee and combined with the suggestions of another reviewer, we added some characterizations and mechanism research, and provided as much as possible a more convincing argumentation and evidence for the rational design of MOFs based Pd/photoredox catalysts.

Lastly, we appreciate very much this referee's patience for our work again.

Independently, this referee suggests considering the following points:

1. Scheme 1 is text-heavy and could be more suited as an "abstract/TOC" image. Particularly panel B is pure text and the Scheme may benefit from enlarging the benefits of this work (panel C) in direct contrast to homogeneous conditions (panel A).

Response: Scheme 1 has been redrawn in the revised manuscript according to the reviewer's suggestions.

2. Statement on line 46 regarding Pd catalyst turnovers often being limited by reoxidation of the Pd⁰ is lacking supporting references.

Response: Reference related to the statement on line 46 has been added in the revised manuscript.

1. Skubi, K. L., Blum, T. R., and Yoon, T. P. Dual Catalysis Strategies in Photochemical Synthesis. *Chem. Rev.* **116**, 10035-10074 (2016).

3. Related to the comment above on originality, it is warranted to include a short discussion and overview on comparable UiO-67 systems with dual-anchored molecular complexes for photocatalytic applications towards the end of the introduction (line 86).

Response: According to the reviewer's suggestion, we added a short discussion at the end of the introduction. The statement is as follows: However, for Pd-catalyzed oxidation reactions, the utilization of MOF based Pd/photoredox catalysts has not been explored. We envision MOFs framework can offer a promising platform for the

regulation of the competitive Pd⁰ aggregation and reoxidation processes. Moreover, the well-defined structures of MOFs can provide facile opportunity to reveal the stepwise electron transfer process between Pd, photocatalyst and O₂, which further gives insight for the elucidation of the Pd reoxidation pathway.

4. The claim in lines 147-149 relating to the ideal diameter(s) for photocatalytic applications is interesting and is likely correct. However, several literature works should be cited to back up the individual claims of i) efficient sensitization, ii) exposure of more active sites, and iii) effective O₂ adsorption.

Response: According to the reviewer's advice, related literatures have been cited to back up the individual claims of i) efficient sensitization (Ref. 49), ii) exposure of more active sites (Ref. 23), and iii) effective O₂ adsorption (Ref. 50).

23. Drake, T., Ji, P. & Lin, W. Site isolation in metal-organic frameworks enables novel transition metal catalysis. *Acc. Chem. Res.* **51**, 2129-2138 (2018).

49. Cao, J., Yang, Z., Xiong, W., Zhou, Y., Peng, Y., Li, X., Zhou, C., Xu, R. & Zhang, Y. One-step synthesis of Co-doped UiO-66 nanoparticle with enhanced removal efficiency of tetracycline: Simultaneous adsorption and photocatalysis. *Chem. Eng. J.* **353**, 126-137 (2018).

50. Choi, J., Yoo, K., Kim, D., Kim, J. & Othman, M. Microporous Mo-UiO-66 metal-organic framework nanoparticles as gas adsorbents. *ACS Appl. Nano Mater.* **4**, 4895-4901 (2021).

5. The EXAFS fit in Figure 2b deviates to the experimental data in several places. A short explanation/clarification would be welcome.

Response: It is noteworthy that the slight deviation of the EXAFS fit of Ir to the experimental data is not uncommon in similar report (*J. Am. Chem. Soc.* **2020**, *142*, 8602; *etc.*). In addition, to further guarantee this EXAFS result, we added the EXAFS result of homogeneous pure Ir complex [Ir(bpy)(ppy)₂](PF₆) in the revised supplementary information (Supplementary Fig. 16a), and it also showed similar result. Meanwhile, the molecular model of the Ir^{III} complex within UiO-67-Ir-PdX₂ (X = OAc, TFA) has been added in the revised SI (Supplementary Fig. 16c) to better

interpret each path in Fig. 2b (in the original manuscript).

6. The described homogeneous catalytic rationale in lines 212-229 is coherent but lacks citations.

Response: Relevant reference has been cited according to the reviewer's suggestion.

15. Piera, J. and Backvall, J. E. Catalytic Oxidation of Organic Substrates by Molecular Oxygen and Hydrogen Peroxide by Multistep Electron Transfer-A Biomimetic Approach. *Angew. Chem. Int. Ed.* **47**, 3506-3523 (2008).

7. In lines 292-294 the post-catalysis crystallinity is discussed and while Figure S14 shows clear retention of reflexes (in-line with the authors' analysis), a short comment/explanation on the distinct change in relative reflex intensities is warranted.

Response: According to the reviewer's suggestion, a short explanation on the change in relative reflex intensities was added in the revised manuscript as follows.

It is worth noting that UiO-67-Ir-Pd(TFA)₂ was stable under the photocatalytic reaction conditions as illustrated by the retention of PXRD pattern for the recovered UiO-67-Ir-Pd(TFA)₂ (Supplementary Fig. 19) as well as the leaching of < 0.5% Pd and < 0.3 % Ir as determined by ICP-MS. The slight changes in the PXRD relative reflex intensities of MOFs catalyst before and after use may be due to the partial structural distortion of the catalyst surface after catalysis. XPS measurements of the catalyst after catalysis showed the oxidation states of Pd (+2) and Ir (+3) remain unchanged (Supplementary Figs. 20 and 21).

8. Table 2 is very interesting and shows a nice scope of tested substrates. For each reaction respectively, similar TONs/Yields are obtained. Can the main source of deactivation be distilled from this? Selectivity issues?

Response: We appreciate the reviewer for the positive comments and suggestion. Indeed, for each reaction respectively, some substrates show similar TONs/Yields. We would like to attribute this phenomenon more to inherent mechanism aspect of the reaction itself. Taking the decarboxylative coupling of allylic alcohols for example, it

was evident that disubstituted benzoic acid facilitates carbon dioxide extrusion process in decarboxylative coupling (*Chem. Eur.J.*, **2014**, *20*, 16680). In addition, the incipient anion which is generated after decarboxylation needs to be efficiently stabilized for further cross-couplings (*Org. Lett.*, **2014**, *16*, 3934). These features make the dimethoxy substituted benzoic acid as the most commonly utilized substrates in Pd-catalyzed decarboxylative coupling (*Org. Lett.*, **2009**, *11*, 2341; *Tetrahedron* **2017**, *73*, 2242; *J. Org. Chem.*, **2016**, *81*, 2521). In this work, the generation of the β -aryl ketones needs selective β -H elimination in the -OH side. The electron-donating dimethoxy benzoic acids could thus facilitate the selective β -H elimination and gave the β -aryl ketones product. Therefore, the dimethoxy benzoic acids are the dominating substrates to determine the yield, and different allylic alcohols with similar dimethoxy benzoic acids show similar TONs/Yields.

9. As stated above, the fs-TAS studies are a welcome addition to the mechanistic analysis. However, from the manuscript it only becomes clear at the end why this was performed on a “control” sample batch specifically including Pd nanoparticles instead of the sample UiO-67-Ir-Pd actually used in catalysis. Including the data for the latter would be interesting to attain a full overview on photophysical behaviour instead of focusing on Ir-Pd⁰ interactions.

Response: We totally understand the consideration of the reviewer. However, as we stated above, these three Pd-catalyzed oxidation reactions are initiated by the interaction between Pd^{II} and substrate, accompanied by the generation of Pd⁰ species. No matter what interaction between Pd^{II} and Ir^{III} in UiO-67-Ir-Pd may happen, the Pd^{II} are the only species to initiate the reaction. The excited [Ir^{III}]* moiety within the framework acts as an oxidant to reoxidize the *in situ* formed Pd⁰ species to Pd^{II}, that is, the electron injection from Pd⁰ to [Ir^{III}]*, which can be monitored by the time-resolved fs-TA spectra and decay constants. While the fs-TA spectroscopic experiment of UiO-67-Ir-PdX₂ cannot reflect the real electron transfer channel and Pd⁰ reoxidation pathway. Thus, the fs-TA experiments of the complex containing Pd⁰ and Ir^{III} species seems reasonable to investigate the reoxidation process of Pd⁰ species.

Anyway, we appreciate the reviewer for the kind suggestions.

10. The same rationale is frequently repeated in a very similar fashion, first the introduction revolves around these points, afterwards in line 212-229 and 316-336 similar formulations for the same arguments are used.

Response: According to the reviewer's advice, the relative statements have been adjusted to be more concisely in the revised manuscript.

11. Figure 1 (b,c,d) quality is poor (at least on this referee's computer) which makes its readability limited.

Response: We readjusted the picture quality in Fig. 1 (b, c, d) (Fig. 2 in the revised manuscript) to make it clear to read.

12. This reviewer believes that "superoxygen" l 431 and 433 should be corrected to "superoxide".

Response: The term "superoxygen" has been corrected to "superoxide" in the revised manuscript.

13. Sentence 433 should be rephrased since superoxide is unlikely to act as an oxidant.

Response: According to the reviewer's advice, this sentence has been deleted, since superoxide is unlikely to act as an oxidant.

14. There is likely to be a structure error in fig 3d

Response: We have revised the structure in Fig. 3d (Fig. 4d in the revised manuscript). We appreciate this reviewer for the careful examination.

We appreciate your consideration of our manuscript, and are looking forward to the acceptance.

Sincerely,

Huanfeng Jiang, Dr. Prof.
Corresponding author

REVIEWERS' COMMENTS

Reviewer #1 (Remarks to the Author):

In the revised manuscript, the authors add new data including XANES, IR, and other controlling experiments. Now all my concerns have been addressed. There are only two small suggestions.

1. In Figure 1, there are too many words and phrases. Please modify Figure 1.
2. The color of the arrow in Figure 2c (for UiO-67-Ir-Pd(OAC)₂) needs to be changed.

Reviewer #2 (Remarks to the Author):

REF: H.Jiang_PhotocatMOF_NatCommun_317681_1

Title: Visible Light-Driven Efficient Palladium Catalyst Turnover in Oxidative Transformations within Confined Frameworks

The revised manuscript from Ren, Jiang, and co-workers addresses the majority of the raised concerns and presents a welcome improvement over the original version.

However, a few points should be considered before the article can be accepted:

1. We appreciate that the authors' provided a detailed response to this reviewer's concerns on conceptual novelty and rational system design in their rebuttal letter and added a short paragraph to the introduction. However, we still feel that the introduction in its current state does not clearly differentiate this work's approach to the state-of-the-art MOF field as merely lines 64-76 are dedicated to this. A few sentences describing the catalyst@UiO-67 literature for photocatalysis (see the first review letter for examples) and then concisely positioning this work in relation to these previous studies is necessary.
2. Directly related to point 1, the claim in line 70-71 that "for Pd-catalyzed oxidation reactions, the utilization of MOF based Pd/photoredox catalysts has never been explored" is not correct. A recent publication uses PCN-222(Pd) for photo-oxidative amine cross-condensation (ACS Sustainable Chem. Eng. 2021, 9, 14405-14415). As this study was not published during this manuscript's writing, this is not the authors' fault. Nevertheless, the claim should be amended and the given publication cited.
3. Figure 1 is still text-heavy with a dark background. A large portion of the text could be moved to the caption to improve readability and clarity.
4. The authors' response to the raised point on similar TONs/yields for the reactions in Table 2 (point 8 in the first review letter) is convincing and well-written. The manuscript would benefit from including some of this discussion to help the reader understand the system's working principles and bottlenecks.

Dear Editors and Reviewers:

We appreciate your recognition concerning our manuscript entitled “Visible Light-Driven Efficient Palladium Catalyst Turnover in Oxidative Transformations within Confined Frameworks”. We have made further corrections according to the reviewer’s advices and the changes are marked with a yellow background in the revised manuscript. The responses to the reviewers are as follows:

Reviewer #1 (Remarks to the Author):

In the revised manuscript, the authors add new data including XANES, IR, and other controlling experiments. Now all my concerns have been addressed. There are only two small suggestions.

1. In Figure 1, there are too many words and phrases. Please modify Figure 1.

Response: According to the reviewer’s suggestion, we modified Figure 1 and moved some words and phrases to the caption to make Figure 1 more concise.

2. The color of the arrow in Figure 2c (for Uio-67-Ir-Pd(OAC)₂) needs to be changed.

Response: We appreciate the careful examination of this reviewer, and the color of the arrow in Figure 2c has been changed.

Reviewer #2 (Remarks to the Author):

The revised manuscript from Ren, Jiang, and co-workers addresses the majority of the raised concerns and presents a welcome improvement over the original version. However, a few points should be considered before the article can be accepted:

1. We appreciate that the authors’ provided a detailed response to this reviewer’s concerns on conceptual novelty and rational system design in their rebuttal letter and added a short paragraph to the introduction. However, we still feel that the

introduction in its current state does not clearly differentiate this work's approach to the state-of-the-art MOF field as merely lines 64-76 are dedicated to this. A few sentences describing the catalyst@UiO-67 literature for photocatalysis (see the first review letter for examples) and then concisely positioning this work in relation to these previous studies is necessary.

Response: According to the reviewer's suggestions, we added further discussion in the revised manuscript. The detailed description is as follows:

The merger of photocatalyst and transition metal catalysts into MOFs have been reported recently, and this elegant methodology has demonstrated their successful applications in photocatalytic water splitting,²⁸⁻³⁰ CO₂ reduction,³¹⁻³³ and organic transformations³⁴⁻³⁷ by promoting electron transfer and stabilizing active intermediates. For example, a Ru-Pt@UIO-67 MOF assembly allows a facile arrangement of the photosensitizer and the reduction catalyst with close spatial proximity to promote the electron transfer between them, and thus leading to a significantly improved hydrogen evolution activity.³⁰ However, to our knowledge, MOF based Pd/photoredox composite has never been explored for Pd catalyst turnover in oxidation reactions.

2. Directly related to point 1, the claim in line 70-71 that "for Pd-catalyzed oxidation reactions, the utilization of MOF based Pd/photoredox catalysts has never been explored" is not correct. A recent publication uses PCN-222(Pd) for photo-oxidative amine cross-condensation (ACS Sustainable Chem. Eng. 2021, 9, 14405-14415). As this study was not published during this manuscript's writing, this is not the authors' fault. Nevertheless, the claim should be amended and the given publication cited.

Response: Firstly, we appreciate this comment from the reviewer. However, after carefully checking out this recent publication (ACS Sustainable Chem. Eng. 2021, 9, 14405-14415), we found that palladium only plays the role in achieving high levels of selectivity and increases the rates of ISC to longer lifetimes of the triplet state. The Pd species valence change and catalyst reoxidation process are not involved in this

publication, which, on the contrary, is the key problem in our work. Thus, technically, this recent publication can be identified as Pd-mediated aerobic photo-oxidative reaction, not traditional Pd-catalyzed oxidation reactions where Pd catalyst turnover is the key process. Anyway, as the reviewer said, the claim “for Pd-catalyzed oxidation reactions, the utilization of MOF based Pd/photoredox catalysts has never been explored” maybe not appropriate, and we revised this statement as “ However, to our knowledge, MOF based Pd/photoredox composite has never been explored for Pd catalyst turnover in oxidation reactions”. We hope this claim would be more scientifically acceptable.

3. Figure 1 is still text-heavy with a dark background. A large portion of the text could be moved to the caption to improve readability and clarity.

Response: According to the reviewer’s advice, we have moved some words and phrases to the caption to make Figure 1 more concise.

4. The authors’ response to the raised point on similar TONs/yields for the reactions in Table 2 (point 8 in the first review letter) is convincing and well-written. The manuscript would benefit from including some of this discussion to help the reader understand the system’s working principles and bottlenecks.

Response: We appreciate the reviewer for the positive comments and suggestion. We have made corresponding statements toward the mechanisms of these oxidation reactions and illustrate the inherent origin of the high efficiency of the MOFs based Pd/photoredox catalysts to these reactions (mechanism discussion in the revised manuscript and Sections 6.1 and 6.2 in Supplementary Information). We hope these discussions would help the reader to better understand these system’s working principles.